# The endonuclease Cue2 cleaves mRNAs at stalled ribosomes during No Go Decay

Karole N D'Orazio[1], Colin Chih-Chien Wu[1], Niladri Sinha[1], Raphael Loll-Krippleber[2], Grant W Brown[2], Rachel Green[1]*

[1]Department of Molecular Biology and Genetics, Howard Hughes Medical Institute, Johns Hopkins University School of Medicine, Baltimore, United States; [2]Donnelly Centre for Cellular and Biomolecular Research, Department of Biochemistry, University of Toronto, Toronto, Canada

**Abstract** Translation of problematic sequences in mRNAs leads to ribosome collisions that trigger a series of quality control events including ribosome rescue, degradation of the stalled nascent polypeptide, and targeting of the mRNA for decay (No Go Decay or NGD). Using a reverse genetic screen in yeast, we identify Cue2 as the conserved endonuclease that is recruited to stalled ribosomes to promote NGD. Ribosome profiling and biochemistry provide strong evidence that Cue2 cleaves mRNA within the A site of the colliding ribosome. We demonstrate that NGD primarily proceeds via Xrn1-mediated exonucleolytic decay and Cue2-mediated endonucleolytic decay normally constitutes a secondary decay pathway. Finally, we show that the Cue2-dependent pathway becomes a major contributor to NGD in cells depleted of factors required for the resolution of stalled ribosome complexes. Together these results provide insights into how multiple decay processes converge to process problematic mRNAs in eukaryotic cells.
DOI: https://doi.org/10.7554/eLife.49117.001

## Introduction

Translation is a highly regulated process in which ribosomes must initiate, elongate, and terminate accurately and efficiently to maintain optimal protein levels. Disruptions in the open-reading frames (ORFs) of mRNAs cause ribosome stalling events, which trigger downstream quality control pathways that carry out ribosome rescue, nascent peptide degradation (via the Ribosome-mediated Quality control Complex or RQC) and mRNA decay (No Go Decay or NGD) (*Brandman and Hegde, 2016*; *Simms et al., 2017a*). Recent work in eukaryotes has revealed that ribosome collisions on problematic mRNAs create a unique interface on the aligned 40S subunits that serves as a substrate for E3 ubiquitin ligases, such as Hel2 and Not4, and the RQC-trigger (RQT) complex, comprised of factors Slh1, Cue2 and Rqt4; together these factors are thought to trigger downstream quality control (*Ferrin and Subramaniam, 2017*; *Garzia et al., 2017*; *Ikeuchi et al., 2019*; *Juszkiewicz et al., 2018*; *Juszkiewicz and Hegde, 2017*; *Matsuo et al., 2017*; *Simms et al., 2017b*; *Sundaramoorthy et al., 2017*). A failure to process such colliding ribosomes and their associated proteotoxic nascent peptide products results in broad cellular distress, made evident by the strong conservation of these pathways (*Balchin et al., 2016*).

Previous genetic screens and biochemistry have identified key factors involved in the recognition of stalling or colliding ribosomes and in targeting the nascent polypeptide to the RQC (*Brandman et al., 2012*; *Kuroha et al., 2010*; *Letzring et al., 2013*). As these earliest screens focused on the identification of factors that stabilized reporter protein expression, factors involved in regulating mRNA decay by NGD remain largely unknown. The hallmark of NGD (*Frischmeyer et al., 2002*; *van Hoof et al., 2002*) is the presence of endonucleolytic cleavage events upstream of the ribosome stalling sequence as first detected by northern analysis (*Doma and*

*For correspondence: ragreen@jhmi.edu

*Parker, 2006*) and more recently by high-resolution ribosome profiling or sequencing approaches (*Arribere and Fire, 2018*; *Guydosh et al., 2017*; *Guydosh and Green, 2017*; *Simms et al., 2017b*). Importantly, these mRNA cut sites depend on ribosome collisions and the consequent polyubiquitination of the 40S subunit by the yeast protein Hel2 (*Ikeuchi et al., 2019*; *Simms et al., 2017b*). Multiple studies in the field collectively position NGD cleavage events within the vicinity of colliding ribosomes (*Arribere and Fire, 2018*; *Guydosh and Green, 2017*; *Guydosh et al., 2017*; *Ibrahim et al., 2018*; *Ikeuchi and Inada, 2016*; *Ikeuchi et al., 2019*; *Simms et al., 2018*; *Simms et al., 2017b*).

Interestingly, an endonuclease has been implicated in a related mRNA decay pathway, Nonsense Mediated Decay (NMD), in metazoans. During NMD, the ribosome translates to a premature stop codon (PTC), where an initial endonucleolytic cleavage event is carried out by a critical PIN-domain containing endonuclease, SMG6 (*Glavan et al., 2006*); sequencing experiments suggest that SMG6 cleavage occurs in the A site of PTC-stalled ribosomes (*Arribere and Fire, 2018*; *Lykke-Andersen et al., 2014*; *Schmidt et al., 2015*). In *C. elegans*, there is evidence that the initial SMG6-mediated endonucleolytic cleavage leads to iterated cleavages upstream of the PTC similar to those characterized for the NGD pathway in yeast (*Arribere and Fire, 2018*). The idea that the various decay pathways may act synergistically is intriguing. Homologs of SMG6 (NMD4/EBS1 in yeast) (*Dehecq et al., 2018*) and other PIN domain-containing proteins, as well as other endonuclease folds, exist throughout Eukarya and anecdotally have been evaluated by the field as potential candidates for functioning in NGD, although there are no reports to indicate that any function in this capacity. The identity of the endonuclease responsible for cleavage in NGD remains unknown.

The presumed utility of endonucleolytic cleavage of a problematic mRNA is to provide access to the mRNA for the canonical exonucleolytic decay machinery that broadly regulates mRNA levels in the cell. In yeast, Xrn1 is the canonical 5' to 3' exonuclease which, after decapping of the mRNA, degrades mRNA from the 5' end; importantly, Xrn1 normally functions co-translationally such that signals from elongating ribosomes might be relevant to its recruitment (*Hu et al., 2009*; *Pelechano et al., 2015*; *Tesina et al., 2019*). Additionally, the exosome is the 3' to 5' exonuclease which, after deadenylation, degrades mRNAs from the 3' end and is recruited by the SKI auxiliary complex consisting of Ski2/Ski3/Ski8 and Ski7 (*Halbach et al., 2013*). While Xrn1-mediated degradation is thought to be the dominant pathway for most general decay in yeast (*Anderson and Parker, 1998*), the exosome has been implicated as critical for many degradation events in the cell including those targeting prematurely polyadenylated mRNAs (these mRNAs are usually referred to as Non-Stop Decay (NSD) targets) (*Frischmeyer et al., 2002*; *Tsuboi et al., 2012*; *van Hoof et al., 2002*). In metazoans, it is less clear what the relative contributions of Xrn1 and the exosome are to the degradation of normal cellular mRNAs. How the endonucleolytic and canonical exonucleolytic decay pathways coordinate their actions on problematic mRNAs remains unknown.

Here, we present a reverse genetic screen in *S. cerevisiae* that identifies Cue2 as the primary endonuclease in NGD. Using ribosome profiling and biochemical assays, we show that Cue2 cleaves mRNAs in the A site of collided ribosomes, and that ribosomes which accumulate at these cleaved sites are rescued by the known ribosome rescue factor Dom34 (*Guydosh and Green, 2014*; *Shoemaker et al., 2010*). We further show that stall-dependent endonucleolytic cleavage represents a relatively minor pathway contributing to the decay of the problematic mRNA reporter used here, while exonucleolytic processing by canonical decay machinery, in particular Xrn1, plays the primary role. Cue2-mediated endonucleolytic cleavage activity is substantially increased in genetic backgrounds lacking the factor Slh1, a known component of the RQT complex (*Matsuo et al., 2017*), suggesting that the relative contribution of this pathway could increase in different environmental conditions. Our final model provides key insights into what happens in cells upon recognition of stalled ribosomes on problematic mRNAs, and reconciles how both endo- and exonucleolytic decay act synergistically to resolve these dead-end translation intermediates.

## Results

### Screening for factors involved in NGD

To identify factors that impact the degradation of mRNAs targeted by NGD, we developed a construct that directly reports on mRNA levels. Previous genetic screens in yeast (*Brandman et al.,*

*2012*; *Kuroha et al., 2010*; *Letzring et al., 2013*) were based on reporters containing a stalling motif in an (ORF). As a result, these screens primarily revealed machinery involved in recognition of stalled ribosomes and in degradation of the nascent polypeptide, but missed factors involved in mRNA decay. In our reporters (*GFP-2A-FLAG-HIS3*), the protein output for the screen (GFP) is decoupled from the stalling motif positioned within *HIS3* by a 2A self-cleaving peptide sequence (*Di Santo et al., 2016*; *Sharma et al., 2012*) (*Figure 1A*). Because GFP is released before the ribosome encounters the stalling sequence within the *HIS3* ORF, its abundance directly reflects the reporter mRNA levels and translation efficiency, independent of the downstream consequences of nascent peptide degradation. These reporters utilize a bidirectional galactose inducible promoter such that an *RFP* transcript is produced from the opposite strand and functions as an internal-control for measurement of general protein synthesis.

The screen utilized two different NGD constructs with stalling motifs inserted into the *HIS3* gene: the first contains 12 CGA codons (NGD-CGA) which are decoded slowly (*Kuroha et al., 2010*; *Letzring et al., 2010*) by the low-copy ICG-tRNA$^{Arg}$ and the second contains 12 AAA codons (NGD-AAA) that mimic the polyA tail and are known to trigger ribosome stalling and mRNA quality control in both yeast and mammalian systems (*Arthur et al., 2015*; *Frischmeyer et al., 2002*; *Garzia et al., 2017*; *Guydosh and Green, 2017*; *Ito-Harashima et al., 2007*; *Juszkiewicz and Hegde, 2017*; *Sundaramoorthy et al., 2017*; *van Hoof et al., 2002*) (*Figure 1A*). The (CGA)$_{12}$ and (AAA)$_{12}$ inserts result in robust threefold and twofold decreases in the GFP/RFP ratio, respectively, compared to the no insert (optimal, OPT) control (*Figure 1B*). Similar changes are also seen in GFP levels by western blot and in full-length mRNA levels by northern blot (*Figure 1—figure supplement 1A*). As previously reported, deletion of the exosome auxiliary factor gene *SKI2* stabilized the 5' decay fragment resulting from endonucleolytic cleavage associated with NGD reporters (*Figure 1—figure supplement 1A*) (*Doma and Parker, 2006*). Finally, stalling during the synthesis of the FLAG-His3 fusion protein in the two stalling reporters (NGD-CGA and NGD-AAA) leads to degradation of the nascent peptide (*Figure 1—figure supplement 1A*, FLAG panel).

We used high-throughput reverse genetic screens and reporter-synthetic genetic array (R-SGA) methods (*Fillingham et al., 2009*; *Tong et al., 2001*) to evaluate the effects of overexpressing annotated genes on GFP expression. We began by crossing strains carrying the control (OPT) and the two different no-go decay reporters (NGD-CGA and NGD-AAA) described in *Figure 1A* into the *S. cerevisiae* overexpression library (*Douglas et al., 2012*; *Giaever et al., 2002*; *Hu et al., 2007*). For each overexpression screen, we isolated diploid strains containing both the overexpression plasmid and our reporter. Selected strains were transferred to galactose-rich plates and the GFP and RFP levels were evaluated by fluorimetry. We plotted the results from the screens individually, comparing Z-scores for the log$_2$(GFP/RFP) signals from each NGD reporter strain to Z-scores from the corresponding strain carrying the OPT reporter (*Figure 1C–1D*). Normalization with RFP intensity was used to eliminate non-specific factors that impact expression of both RFP and GFP.

The overexpression screen revealed a set of candidate genes that modulate GFP levels for the NGD reporters relative to the OPT reporter (*Figure 1C–1D* and *Figure 1—source data 1*). Broadly, we see a stronger overlap in candidate overexpression genes among the NGD reporters than for either NGD reporter compared to the OPT reporter (*Figure 1—figure supplement 1B*). By far, the strongest outlier by Z-score causing reduced GFP expression for both NGD reporters, without affecting the OPT reporter, resulted from overexpression of the gene *CUE2* (*Figure 1C–1D*). Flow cytometry and northern analysis confirm that NGD-CGA reporter mRNA levels are substantially reduced upon *CUE2* overexpression whereas the control RFP transcript is not affected (*Figure 1E* and *Figure 1—figure supplement 1C*).

## *CUE2* domain structure and homology modeling

The domain structure of Cue2 reveals it to be a promising candidate for the missing endonuclease for NGD. Cue2 contains two conserved CUE (coupling of ubiquitin to ER degradation) ubiquitin-binding domains at the N-terminus (*Kang et al., 2003*), followed by two putative ubiquitin-binding domains (UBA* (ubiquitin-associated domain) and CUE* domain, respectively), and an SMR (small MutS-related) hydrolase domain at the C-terminus (*Figure 2A*). We performed alignments of the CUE and SMR domains of Cue2 using structure-based alignment tools (*Figure 2—figure supplement 1A*; *Figure 2—figure supplement 1B*) and found putative homologs in various kingdoms of life, including the human NEDD4-binding protein 2, N4BP2. While the SMR domain of MutS family

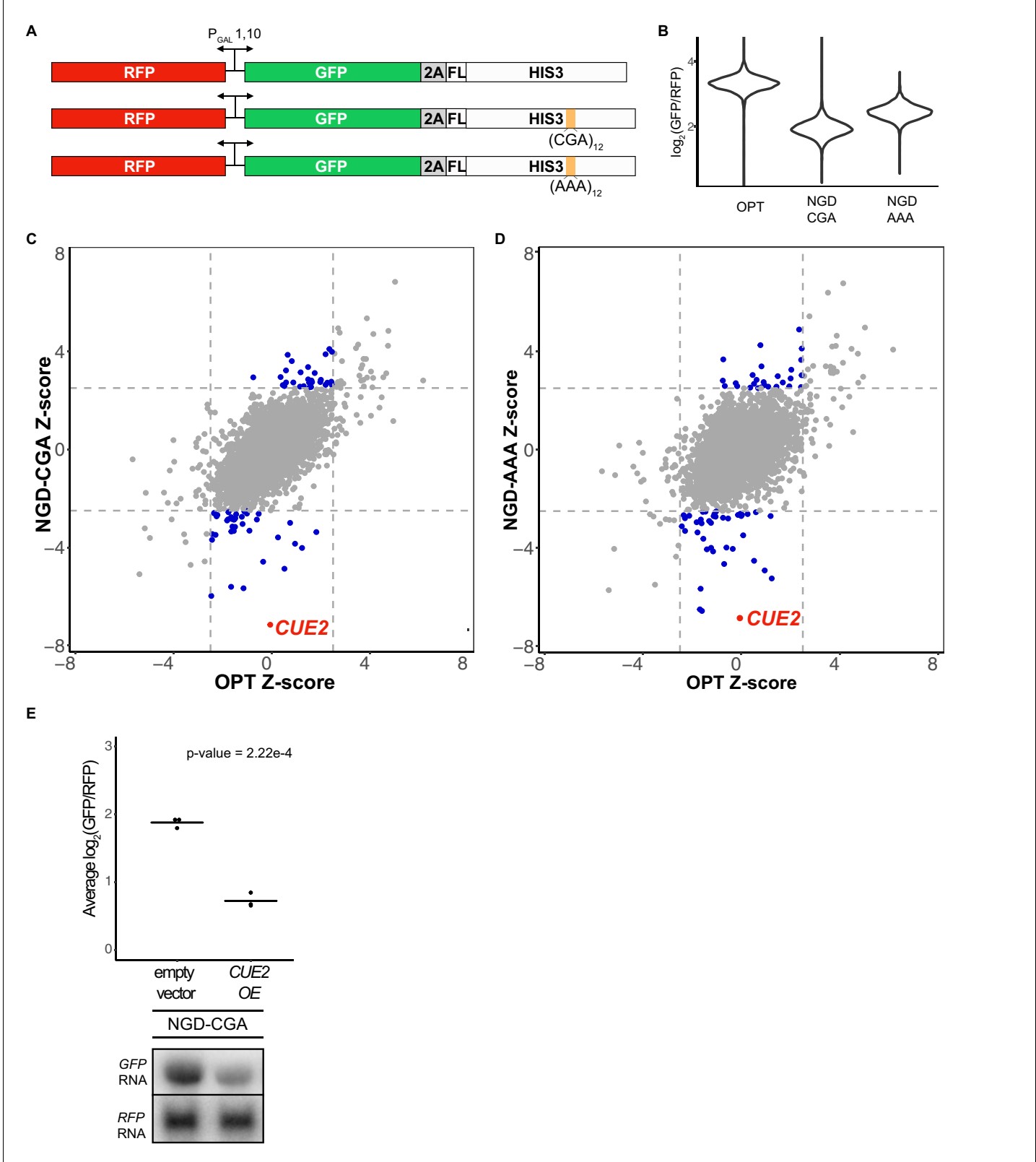

**Figure 1.** Yeast overexpression screens identify a novel factor involved in NGD. (A) Schematic of reporters used in genetic screens. Top is 'OPT'; middle is 'NGD-CGA'; bottom is 'NGD-AAA'. (B) Normalized reporter GFP levels. Violin plots show flow cytometry data from >4000 WT cells containing the indicated reporter. (C–D) Plots of Z-scores from overexpression screens comparing the NGD-CGA and NGD-AAA reporters to OPT. Z-scores reflecting the significance of $\log_2(GFP/RFP)$ values from each strain are plotted against each other for the two reporters. Dashed lines represent cutoffs

*Figure 1 continued on next page*

*Figure 1 continued*

at a Z-score greater than 2.5 or less than −2.5 for each reporter. Blue dots represent overexpression strains that have a Z-score value outside the cut-off for the NGD reporters, but not the OPT. Red dots identify the *CUE2* overexpression strain. (**E**) Validation data for overexpression screen candidate *CUE2*. Top, three averages from individual flow cytometry experiments are plotted for the NGD-CGA reporter strain without *CUE2* overexpression (left), and with *CUE2* overexpression (right). Bottom, northern blots of steady state mRNA levels for the same strains. See *Figure 1—figure supplement 1*.
DOI: https://doi.org/10.7554/eLife.49117.002

The following source data and figure supplement are available for figure 1:

**Source data 1.** Overexpression screen results.
DOI: https://doi.org/10.7554/eLife.49117.004
**Figure supplement 1.** Validation of NGD reporters and screen results.
DOI: https://doi.org/10.7554/eLife.49117.003

enzymes canonically functions as a DNA-nicking hydrolase (*Fukui and Kuramitsu, 2011*), the SMR domain of SOT1 in plants exhibits RNA endonuclease activity (*Zhou et al., 2017*). Additionally, SMR domains show structural similarity to bacterial RNase E (*Fukui and Kuramitsu, 2011*). Alignments of the SMR domains of these and other proteins enabled us to identify conserved residues (*Figure 2—figure supplement 1B*) some of which are known to be critical for RNA endonuclease activity in the plant enzyme (*Zhou et al., 2017*).

Heuristic searches of the structurally defined SMR domain of the mammalian homolog of Cue2 (N4BP2) (*Diercks et al., 2008*) against known structures in the Protein Data Bank found it to be structurally homologous to the C-terminal domain (CTD) of bacterial Initiation Factor 3 (*Biou et al., 1995*) (*Figure 2B* and *Figure 2—figure supplement 1C–D*). During initiation in bacteria, IF3 binds to the 30S pre-initiation complex (PIC) and helps position the initiator tRNA at the AUG start codon; the CTD of IF3 binds in close proximity to the P and A sites of the small subunit, closely approaching the mRNA channel (*Hussain et al., 2016*). We aligned the Cue2-SMR homology model with the structure of the IF3-CTD on the ribosome and observed that conserved residues D348, H350, and R402 are positioned along the mRNA channel in this model (*Figure 2C–2D* and *Figure 2—figure supplement 1E–F*) and thus represent potential residues critical for endonucleolytic cleavage activity; additionally, R402 had previously been implicated in the RNA cleavage activity of other SMR domains (*Zhou et al., 2017*).

## Characterizing roles of *CUE2* in NGD in vivo

We performed several different experiments to ask if these conserved residues are necessary for Cue2 function. First, we generated mutations in HA-tagged Cue2 and performed flow cytometry on the NGD-CGA reporter under overexpression conditions for the different Cue2 variants. We find that individually mutating the conserved residues D348, H350, and R402 to alanine (A) causes a modest increase in GFP expression relative to *CUE2* WT, whereas the R402K mutation (which maintains a positively charged amino acid) has no discernible effect (*Figure 2E*). Mutating residues 348 through 350, which includes the conserved residues D348 and H350, nearly restores GFP reporter signal in this assay (*Figure 2E*). Importantly, each of these variants is expressed at similar levels (*Figure 2—figure supplement 1G*). These data suggest a potential role for residues D348, H350, and R402 in the SMR domain of Cue2 in reducing levels of problematic mRNAs.

We next asked if Cue2 is necessary for the previously documented endonucleolytic cleavage of the NGD reporter transcripts. In order to visualize the mRNA fragments resulting from cleavage, we deleted either the 3' - 5' mRNA decay auxiliary factor, *SKI2*, or the major 5' - 3' exonuclease, *XRN1*, to stabilize the 5'- and 3'-fragments, respectively (*Doma and Parker, 2006*) (*Figure 2F*; lane 1 and *Figure 2G*; lane 1). Upon deletion of *CUE2* in the appropriate yeast background, we see that both of these decay intermediates disappear (*Figure 2F*; lane 2 and *Figure 2G*; lane 2).

In order to further investigate the role of the specific amino acids proposed to be catalytically critical, we modified *CUE2* at its endogenous chromosomal locus. We added an HA tag to allow us to follow native *CUE2* protein levels and generated mutations in *CUE2* in a *ski2Δ* background. As a first test, we deleted the C-terminal portion of *CUE2* comprising the SMR domain and revealed a complete loss of endonucleolytic cleavage (*Figure 2H*, lane 3). To further refine this analysis, we made combined mutations in *CUE2* at D348 and H350 and observed a complete abolishment of endonucleolytic cleavage activity (*Figure 2H*, lane 4). Western analysis confirmed that protein expression

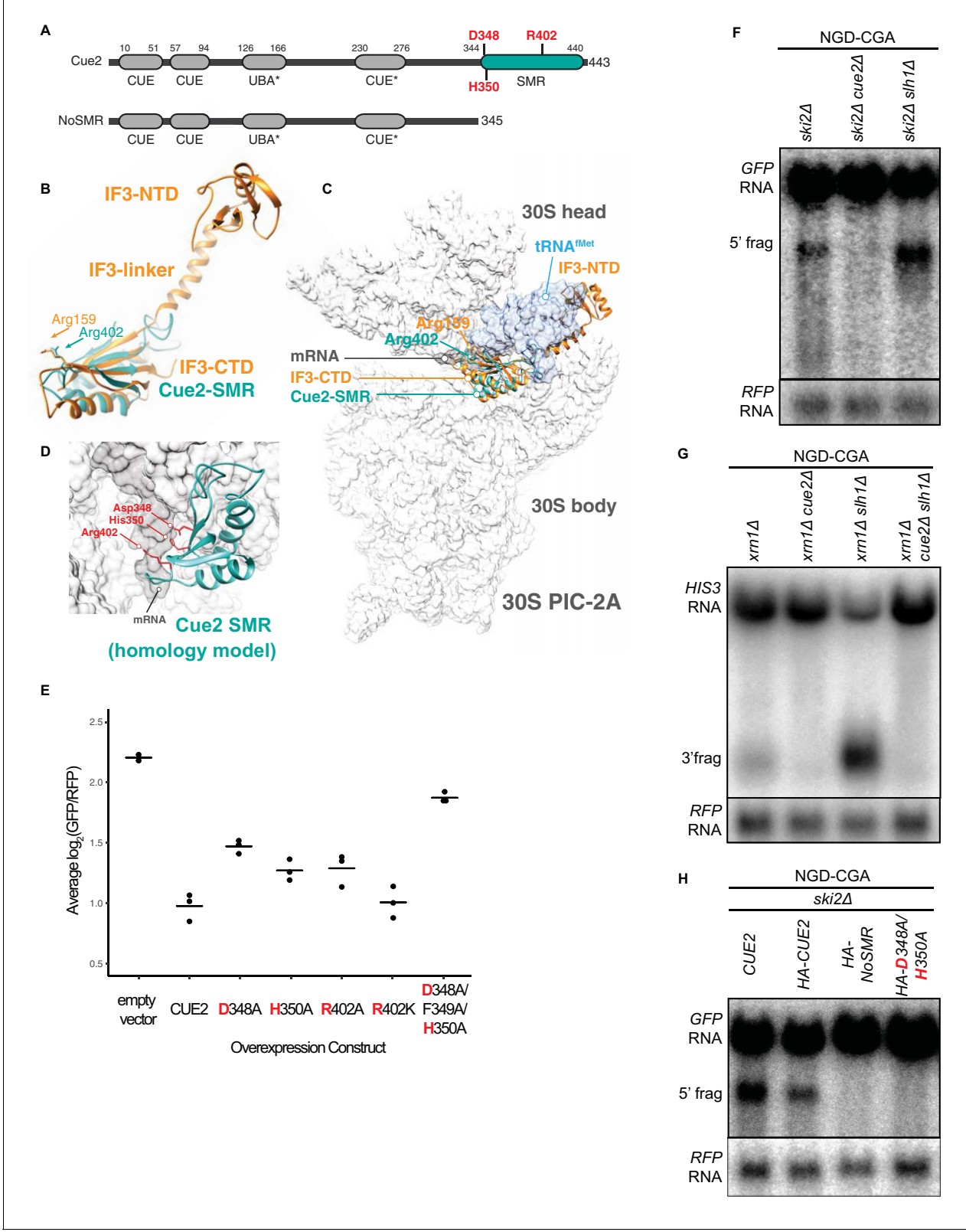

**Figure 2.** Cue2 domain structure and in vivo characterization of Cue2 in NGD. (**A**) Domain organization of *S. cerevisiae* Cue2, with the 'NoSMR' mutant schematic below the WT Cue2. Putative ubiquitin binding domains indicated by asterisks. (**B**) Superimposition of homology model of Cue2-SMR domain (cyan) (templated on human N4BP2-SMR; PDB: 2VKC [*Diercks et al., 2008*]) on full-length IF3 (orange, PDB: 5LMQ [*Hussain et al., 2016*]). Positions of Arg-159 of IF3-CTD and Arg-402 of Cue2-SMR are indicated. (**C**) Putative positioning of Cue2-SMR in the context of IF3 and tRNA[fMet]

*Figure 2 continued on next page*

*Figure 2 continued*
bound to the small ribosome subunit. 30S PIC-2A (as described in *Hussain et al., 2016*) is light gray; mRNA is dark gray; tRNA^fMet is blue; IF3 is orange; Cue2-SMR is cyan. (D) Cue2-SMR homology model in the same orientation as seen in *Figure 2—figure supplement 1F* showing putative positions of Asp-348, His-350 and Arg-402 with 30S pre-initiation complex (PIC) with tRNA^fMet density subtracted (PDB: 5LMQ, State 2A [*Hussain et al., 2016*]). (E) Mutational analysis of the Cue2-SMR domain. Cue2 WT and mutants are all HA tagged at the N-terminus. Three averages for individual sets of log$_2$(GFP/RFP) values of flow cytometry experiments for each indicated overexpression construct are shown. p-Values for the mutant *HA-CUE2* as compared to *WT HA-CUE2* are 0.007195, 0.02602, 0.03857, 0.7788, and 0.002127 for D348A, H350A, R402A, R402K, D348A/F349A/H350A, respectively. (F–G) Northern blot analysis of the indicated strains, with full-length mRNA and the 5' (F) and 3' (G) fragments labeled; RFP probed for normalization. (H) Northern blot analysis of 5' NGD-CGA cleavage fragments in the *ski2Δ* strain with the indicated mutations made at the endogenous *CUE2* locus; RFP probed for normalization. See *Figure 2—figure supplement 1*.
DOI: https://doi.org/10.7554/eLife.49117.005
The following figure supplement is available for figure 2:

**Figure supplement 1.** Characterizing Cue2 domains and confirming mutant *in vivo* expression levels.
DOI: https://doi.org/10.7554/eLife.49117.006

levels from the CUE2 locus are equivalent for these different protein variants (*Figure 2—figure supplement 1H*). These data provide strong support for a hydrolytic role for Cue2 in the endonucleolytic cleavage of problematic mRNAs in the NGD pathway.

## Contribution of Cue2 to NGD is increased in specific genetic backgrounds

Surprisingly, deletion of *CUE2* in the WT, *ski2Δ* or *xrn1Δ* background does not demonstrably impact GFP or steady-state full-length mRNA levels (*Figure 3A–3B*, *Figure 2F*, or *Figure 2G*, respectively) suggesting that the endonucleolytic decay pathway is not responsible for the majority of the three-fold loss in mRNA levels observed for this NGD reporter (relative to OPT, *Figure 1—figure supplement 1A*). However, we see that deleting *XRN1* alone greatly restores NGD-CGA reporter mRNA levels while deleting *SKI2* alone has very little effect (*Figure 3A–3B*). These results together suggest that the majority of mRNA degradation observed for the NGD-CGA reporter is mediated by canonical decay pathways, mostly through Xrn1.

A key insight into the regulation of decay by these distinct mRNA decay pathways (endo- and exonucleolytic) came from examining decay intermediates in strains lacking the factor Slh1. Slh1 is a member of the Ribosome Quality control Trigger (RQT) complex that was previously implicated in regulating NGD (*Ikeuchi et al., 2019*; *Matsuo et al., 2017*). This large protein (~200 kDa) is homologous to helicases implicated in RNA decay and splicing including Ski2 and Brr2, respectively (*Johnson and Jackson, 2013*). Slh1 contains two RecA-like helicase cassettes with DEAD box motifs and a K to R mutation in the first RecA motif disrupts targeting of the nascent peptide on problematic mRNAs to proteolytic decay by the RQC. Based on these observations, Slh1 has been proposed to directly or indirectly dissociate stalled ribosomes via its ATPase activity and to thereby target them for the RQC (*Matsuo et al., 2017*).

We found that in the *ski2Δ slh1Δ* and *xrn1Δ slh1Δ* strains, we see a sizeable build-up of endonucleolytically cleaved mRNA fragments (*Figure 2F*; lane 3 and *Figure 2G*; lane 3). Consistent with the recent study by *Ikeuchi et al. (2019)*, we also see smearing of the signal representative of shorter 5' and longer 3' fragments due to additional cleavage events occurring upstream of the primary stall site in the absence of *SLH1*. Importantly, we also observe a robust decrease in full-length mRNA levels upon deletion of *SLH1* (*Figure 2F*; lane 3 and *Figure 2G*; lane 3). These data suggest a model wherein Slh1 activity somehow competes with Cue2-mediated endonucleolytic cleavage. To ask whether the increased endonucleolytic cleavage seen in the *SLH1* deletion background is the result of Cue2 activity, we deleted *CUE2* in the *xrn1Δ slh1Δ* background and observed a complete loss of the decay intermediate and a restoration of full-length mRNA levels (*Figure 2G*; lane 4). These data provide strong evidence that the action of Slh1 negatively regulates the assembly of a potent Cue2 substrate though the molecular mechanism for this regulation is not defined.

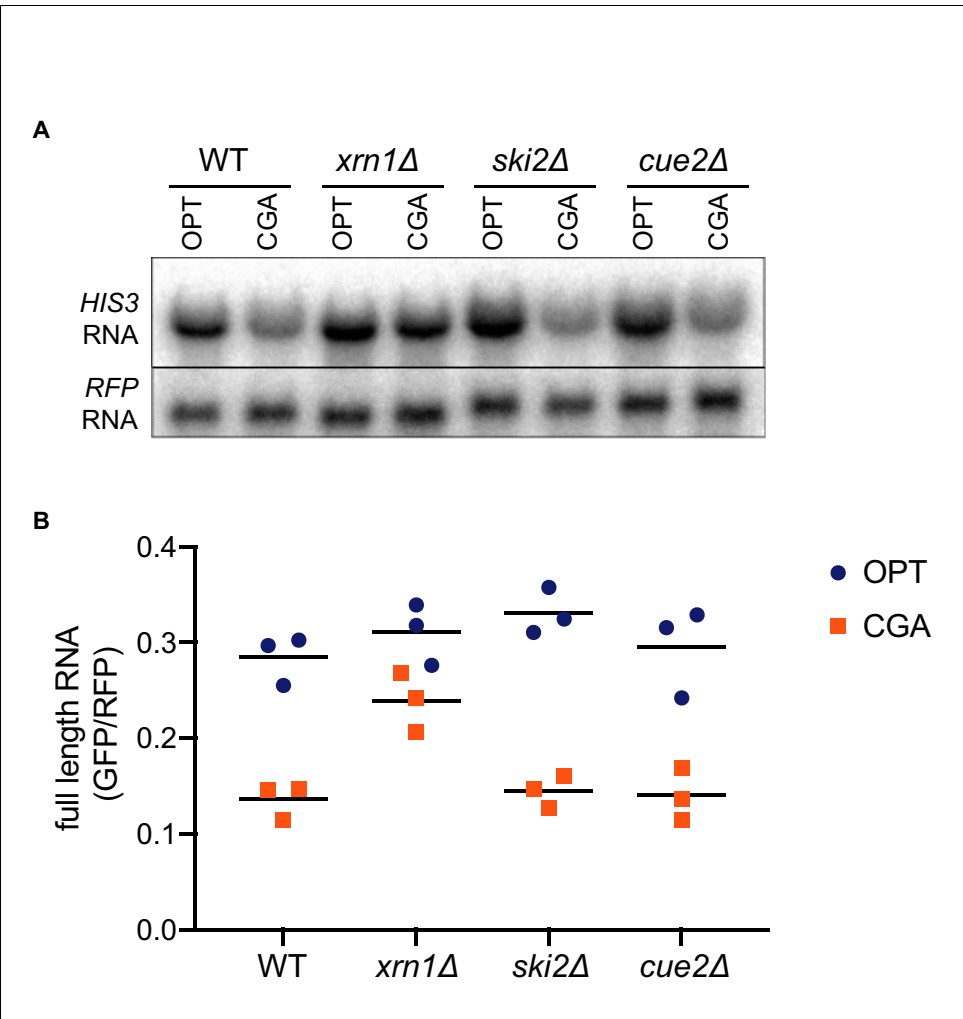

**Figure 3.** Canonical exonucleolytic decay by Xrn1 is the major contributor to NGD. (**A**) Northern blot analysis of full length OPT and NGD-CGA reporter levels in the indicated strains. (**B**) Quantitated northern blot signals (in triplicate) from full length GFP/RFP mRNA levels for the indicated strains are plotted.

DOI: https://doi.org/10.7554/eLife.49117.007

## Ribosome profiling provides high-resolution view of Cue2 cleavage sites

We next utilized ribosome profiling to assess the role and specificity of Cue2 in endonucleolytic cleavage of problematic mRNAs. Ribosome profiling was performed in various genetic backgrounds where the NGD-CGA reporter construct was included to follow the process of NGD on a well-defined mRNA substrate. The approach benefits from our ability to distinguish three distinct sizes of ribosome-protected mRNA fragment (RPF) that correspond to three different states of the ribosome during translation. 16 nucleotide (nt) RPFs correspond to ribosomes that have translated to the truncated end of an mRNA in the cell, usually an mRNA that has been endonucleolytically processed; these 16 nt RFPs are enriched in a *dom34Δ* strain since these are established targets for Dom34-mediated ribosome rescue (*Arribere and Fire, 2018*; *Guydosh and Green, 2017*; *Guydosh and Green, 2014*; *Guydosh et al., 2017*). 21 and 28 nt RPFs are derived from ribosomes on intact mRNA, with the two sizes corresponding to different conformational states of ribosomes in the elongation cycle (*Lareau et al., 2014*; *Wu et al., 2019b*). In our modified ribosome profiling library preparation (*Wu et al., 2019b*) 21 nt RPFs represent classical/decoding ribosomes waiting for the next aminoacyl-tRNA whereas the 28 nt RPFs primarily represent ribosomes in a rotated/pre-translocation state. Throughout this study, we performed ribosome profiling in a *ski2Δ* background to prevent

3′–5′ exonucleolytic degradation by the exosome and enable detection of mRNA decay intermediates. We note that deleting *SKI2* generally stabilizes prematurely polyadenylated and truncated mRNAs in the cell, as seen previously (*Tsuboi et al., 2012*; *van Hoof et al., 2002*); therefore, use of this background likely enhanced Cue2 cleavage on these mRNA substrates.

In a first set of experiments, we compare the distributions of RPFs in *ski2Δ* strains to strains additionally lacking *DOM34*, or *DOM34* and *CUE2*; in each case, we separately consider the different RPF sizes (16, 21 and 28, *Wu et al., 2019a* copy archived at https://github.com/elifesciences-publications/Cue2eLife) to determine the state of the ribosome on these sequences. In *ski2Δ*, we observe a clear accumulation of ribosome density at the $(CGA)_{12}$ stall site (in the 21 nt RPF track) in our reporter (*Figure 4A*) consistent with ribosome stalling at this site. Looking first within the actual $(CGA)_{12}$ codon tract, we see a quadruplet of 21 nt RPF peaks from the second to the fifth CGA codon (*Figure 4A*, middle panel). Because of difficulties in mapping monosome reads to repetitive sequences such as $(CGA)_{12}$, we also isolated longer RPFs from a minor (<5% of total reads on reporter) population of stacked disome species (*Guydosh and Green, 2014*) and show that although some ribosomes move into the subsequent CGA codons, the principal ribosome accumulation occurs between the second and the fifth codon (*Figure 4—figure supplement 1*). While 21 nt RFPs are enriched on CGA codons, 28 nt RPFs are not (*Figure 4A*, compare middle and bottom panels), indicating that most ribosomes found at the CGA codons are waiting to decode the next aminoacyl-tRNA; these observations are consistent with the fact that CGA codons are poorly decoded (*Letzring et al., 2010*). The 21 nt RPF quadruplet at the CGA cluster represents the stalled 'lead' ribosome in its classical (unrotated) conformation, which is consistent with recent cryo-EM structures of collided ribosomes (*Ikeuchi et al., 2019*; *Juszkiewicz et al., 2018*). About 30 nts upstream of these lead ribosomes, we observe the accumulation of 28 nt RPFs that correspond to colliding ribosomes in a rotated state (*Figure 4A*, bottom panel), also in agreement with the ribosome states recently reported in cryo-EM structures of collided disomes. In the *dom34Δ ski2Δ* strain (where 16 nt RPFs are stabilized), we observe a strong quadruplet of 16 nt RPF peaks exactly 30 nts upstream of the quadruplet of 21 nt RPFs (*Figure 4B*), indicative of endonucleolytic cleavage events occurring upstream of the $(CGA)_{12}$–stalled ribosomes. The precise 30 nt distance between the 16 nt RPFs and the downstream 21 nt RPFs is consistent with the measured distance between the A sites of two colliding ribosomes.

To address the role of Cue2 as the endonuclease, we examined the same collection of RPFs in the *cue2Δ dom34Δ ski2Δ* strain. In this background, the 16 nt RPFs indicative of endonucleolytic cleavage are dramatically reduced in abundance while the distributions of 21 and 28 nt RPFs are largely unaffected (*Figure 4C*). These data argue that endonucleolytic cleavage occurs precisely in the A site of the collided, rotated ribosome and requires Cue2. A high-resolution view of the site of cleavage within the A site codon is found in the model in *Figure 4G*. These observations are broadly consistent with recent RACE analyses (*Ikeuchi et al., 2019*) and with our structural homology modeling placing Cue2 in the decoding center of the ribosome (*Figure 2C and 2D*).

Based on our observations that deletion of Slh1 increases utilization of Cue2 for processing of problematic mRNAs, we next examined the distribution of RPFs on the NGD-CGA reporter in yeast strains lacking *SLH1*. In the *slh1Δ ski2Δ* strain, the pattern of RPF distributions is broadly similar to what we observed in the *ski2Δ* strain (*Figure 4D* compared to 4A): 21 nt RPFs accumulate on the second through fifth CGA codons and 28 nt RPFs accumulate 30 nts behind the lead ribosome; we also see modest accumulation of RPFs downstream of the stalling sequence as previously reported (*Figure 4D*) (*Sitron et al., 2017*). *SLH1* deletion does not stabilize 16 nt RPFs, and therefore, likely does not act on cleaved Cue2-products (*Figure 4D* compared to 4A). When we delete *DOM34* in the *slh1Δ ski2Δ* background (*slh1Δ dom34Δ ski2Δ*), we see a dramatic accumulation of 16 nt RPF peaks (again, as a quadruplet, although with an altered relative distribution) behind the lead ribosomes (*Figure 4E* compared to 4B, top panels, note the larger y-axis scale in *Figure 4E and 4F* compared to 4B and 4C). These data are consistent with the substantial increase in cleavage we see via northern blot (*Figure 2F and 2G*) and previous findings that deletion of the RQT complex causes somewhat distinct cleavage events (*Ikeuchi et al., 2019*). Importantly, deletion of *CUE2* in the *slh1Δ dom34Δ ski2Δ* background leads to a near complete loss of the 16 nt RPFs (*Figure 4F*, top panel) and an enrichment of 21 and 28 nt RPFs.

Lastly, consistent with earlier studies showing that endonucleolytic cleavage events associated with NGD depend on the E3 ligase Hel2 (*Ikeuchi et al., 2019*), we show that deletion of *HEL2* in

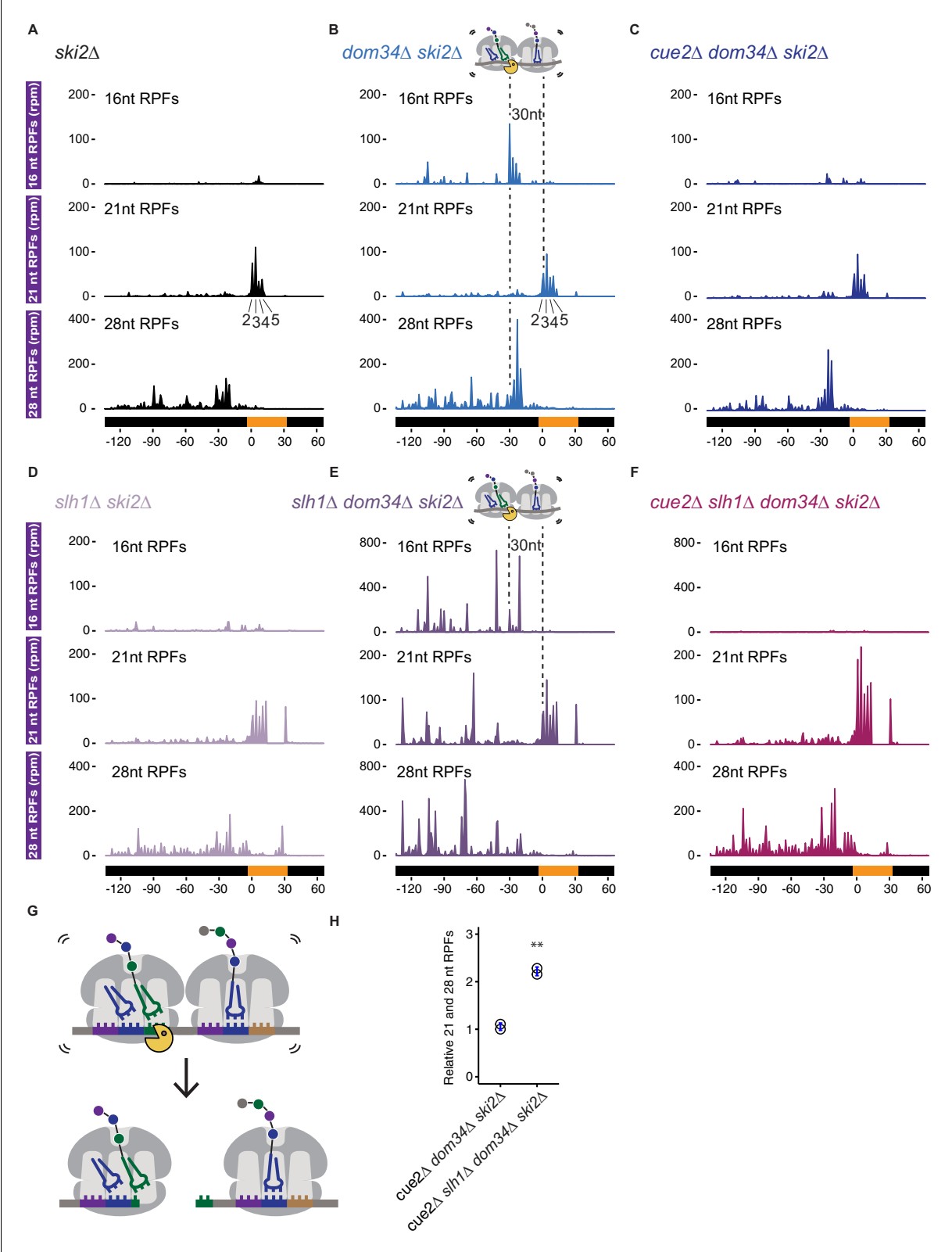

**Figure 4.** Ribosome profiling analysis of NGD on reporter mRNAs in various genetic backgrounds. (A–F) 16, 21 and 28 nt RPFs mapped to NGD-CGA reporter in *ski2Δ* (A), *dom34Δ ski2Δ* (B), *cue2Δ dom34Δ ski2Δ* (C), *slh1Δ ski2Δ* (D), *slh1Δ dom34Δ ski2Δ* (E), and *cue2Δ slh1Δ dom34Δ ski2Δ* (F) strains with schematic depicting Cue2-mediated cleavage in the A sites of collided ribosomes. (G) Schematic of the precise Cue2 cleavage location on the mRNA, relative to collided ribosomes. (H) Comparison of the combined 21 and 28 nt ribosome occupancies on GFP, from 300 nt upstream of the (CGA)₁₂ to

*Figure 4 continued on next page*

*Figure 4 continued*

the end of the (CGA)$_{12}$ sequence, normalized to RFP for the indicated strains (n = 2). p-value from Student's t-test is indicated by asterisks. **, p<0.01.
See *Figure 4—figure supplements 1–2*.
DOI: https://doi.org/10.7554/eLife.49117.008
The following figure supplements are available for figure 4:

**Figure supplement 1.** Disome and monosome footprints are consistent in mapping the leading ribosome.
DOI: https://doi.org/10.7554/eLife.49117.009
**Figure supplement 2.** Hel2 is required for endonucleolytic cleavage at stall sites.
DOI: https://doi.org/10.7554/eLife.49117.010

both *dom34Δ ski2Δ* and *slh1Δ dom34Δ ski2Δ* backgrounds leads to a complete loss of 16 nt RPFs (*Figure 4—figure supplement 2A* and f, respectively).

## In vitro reconstitution of Cue2 cleavage on isolated colliding ribosomes

To test if Cue2 cleaves mRNA directly, we performed in vitro cleavage assays with the heterologously expressed SMR domain (and R402A mutant) of Cue2 and purified colliding ribosomes. To isolate the colliding ribosomes, cell-wide ribosome collision events were induced using low-dose cycloheximide treatment of growing yeast (*Simms et al., 2017b*). We both optimized the yield and ensured that the purified Cue2-SMR domain was the only source of Cue2 in our experiment by isolating ribosomes from the *cue2Δ slh1Δ dom34Δ ski2Δ* strain; these cells also carried and expressed the NGD-CGA reporter. Since collided ribosomes are resistant to general nucleases (*Juszkiewicz et al., 2018*), MNase digestion of lysates collapses elongating ribosomes from polysomes to monosomes but spares those with closely packed (nuclease resistant) ribosomes (*Figure 5—figure supplement 1*). As anticipated, only cells treated with low doses of cycloheximide yielded a substantial population of nuclease resistant, or collided ribosomes after MNase digestion (*Figure 5—figure supplement 1*, black trace). We isolated nuclease-resistant trisomes from a sucrose gradient to serve as the substrate for our in vitro reconstituted cleavage experiment.

We purified the isolated SMR domain of Cue2 based on our in vivo evidence that this domain represents the functional endonuclease portion of the protein. We added the purified SMR domain of Cue2 to the isolated nuclease-resistant trisomes and resolved the products of the reaction on a sucrose gradient that optimally resolves trisomes from monosomes and disomes. While we initially isolated trisomes, it is clear that the 'untreated' sample (black trace) contains trisomes, as well as disomes and monosomes; this may be the result of cross-contamination of those peaks during purification or the instability of the trisome complex. Nevertheless, on addition of Cue2-SMR (WT) (pink trace), we observe a substantial loss of trisomes and a corresponding increase in monosomes and disomes (*Figure 5A*), while on addition of the Cue2 SMR-R402A mutant (orange trace), we observe a more modest decrease in trisomes (*Figure 5A*). These results provide initial in vitro evidence that the SMR (hydrolase) domain of Cue2 cleaves the mRNA within a stack of colliding ribosomes.

To extend these observations, we mapped more precisely the in vitro cleavage sites of the SMR domain by sequencing the ribosome-protected mRNA fragments from the untreated and SMR-treated samples. Because the strain used to prepare the cycloheximide-induced colliding ribosomes also expressed the NGD-CGA reporter, we anticipated that within our population of bulk colliding ribosomes would be a sub-population of colliding ribosomes on our reporter, also subject to cleavage by the recombinant SMR domain. We performed RNA-seq by standard means from the trisomes treated with or without the Cue2 SMR domain and aligned the 3' ends of the RNA reads to the reporter sequence. In the sample treated with the wild-type Cue2-SMR domain, we see a striking accumulation of 3' fragment ends that maps precisely where the strong cleavage sites were reported (in the A site of the colliding ribosome) in the ribosome profiling data for the same strain (*slh1Δ dom34Δ sk2iΔ*); reassuringly, these cleavage events are not seen in the sequencing data for the Cue2 SMR-R402A mutant or for the no enzyme control (*Figure 5B*). The remarkable agreement between the in vivo-based ribosome profiling data and the in vitro-based cleavage data provides strong support for the identity of Cue2 as the endonuclease involved in NGD.

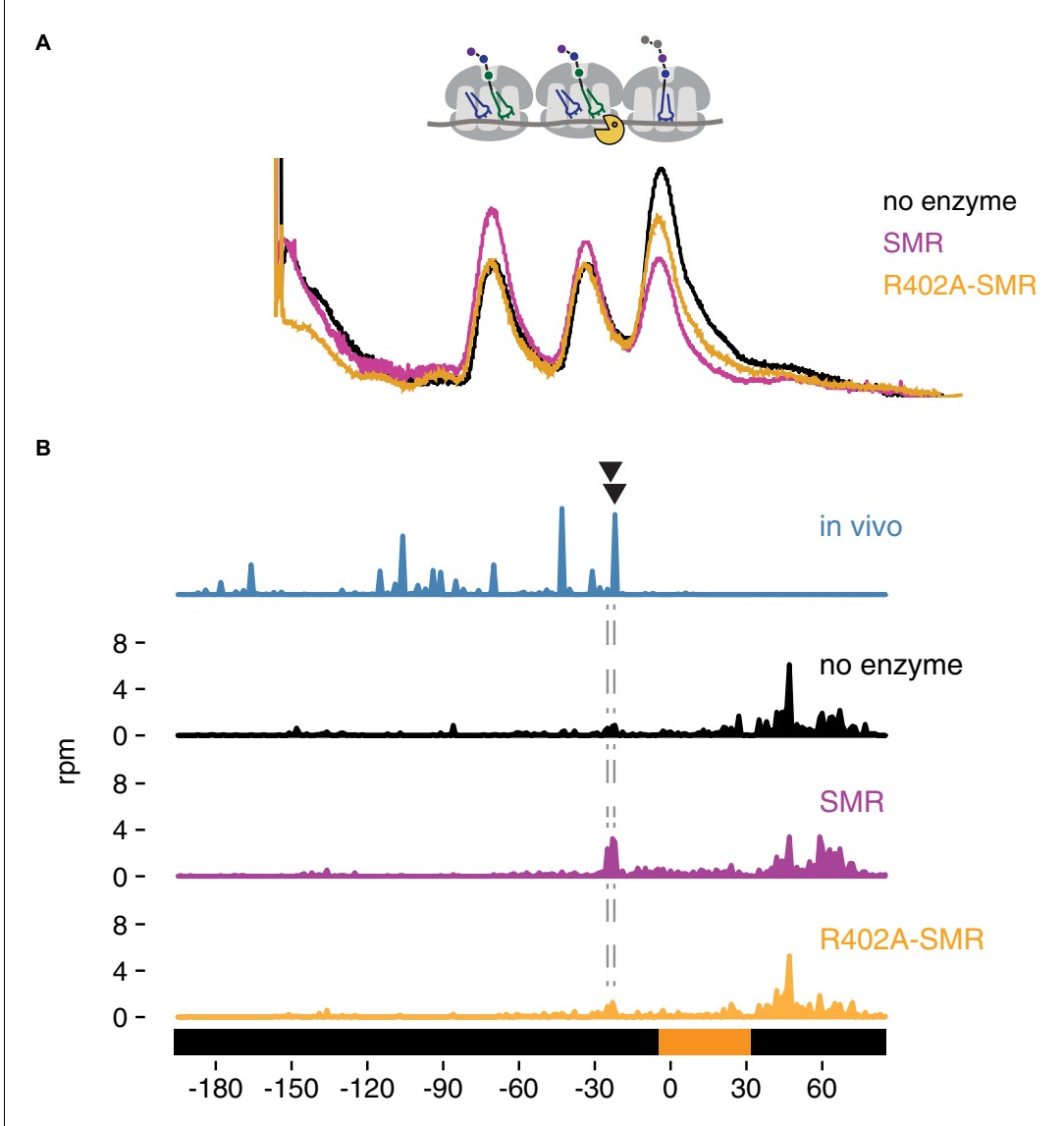

**Figure 5.** In vitro cleavage of colliding ribosomes by purified Cue2-SMR. (**A**) Absorbance at 260 nm (y-axis) for sucrose gradient of nuclease resistant trisomes treated with no enzyme (black), the SMR domain of Cue2 (pink), or the SMR-R402A mutant Cue2 (orange). (**B**) In vivo 16 nt RPFs from the *slh1Δ dom34Δ ski2Δ* strain (cyan) aligned relative to the 60–65 nt RPFs from the combined fractions (mono-, di- and trisomes) treated with no enzyme (black), SMR domain (pink), or R402-SMR (orange) in vitro. Arrowheads indicate the positions where the in vitro and the in vivo cleavages coincide. See *Figure 5—figure supplement 1*.

DOI: https://doi.org/10.7554/eLife.49117.011

The following figure supplement is available for figure 5:

**Figure supplement 1.** Cycloheximide-induced cells give MNase-resistant trisome peak.

DOI: https://doi.org/10.7554/eLife.49117.012

## Ribosome profiling provides evidence that Slh1 inhibits ribosome accumulation on problematic mRNAs

Our analyses (northern blots and profiling) provide strong evidence that deletion of *SLH1* in yeast results in increased levels of Cue2 cleavage on our NGD-CGA reporter. Multiple molecular mechanisms could be responsible for this outcome. For example, a simple model might involve direct competition on the colliding ribosomes wherein Slh1 binding simply blocks Cue2 from binding to the same site. Another model could be that Slh1 functions to clear colliding ribosomes, dissociating large and small subunits from one another, and thus allowing the released peptidyl-tRNA:60S

complex to be targeted for RQC (*Matsuo et al., 2017*). Lastly, earlier studies have argued that Slh1 functions as a translational repressor, though the dependence of RQC activity on Slh1 function argues against this model (*Searfoss et al., 2001*).

We used ribosome profiling to directly assess ribosome occupancy on our problematic mRNA sequence as a function of Slh1 activity. As described above, when we compared 16 nt RPF accumulation on the NGD-CGA reporter in a *ski2Δ* or *slh1Δ ski2Δ* strain (*Figure 4A and 4C*) we saw no differential accumulation of these RPFs that might be indicative of Slh1 function on ribosomes trapped on truncated mRNAs. However, when we compared RPFs on the uncleaved mRNAs (i.e. the 21 and 28 nt RPFs) in the *cue2Δ dom34Δ ski2Δ* and *slh1Δ cue2Δ dom34Δ ski2Δ* strains, we observed that *SLH1* deletion results in a substantial build-up of RPFs both at the stalling site (21 nt RPFs) (compare middle panels, *Figure 4C and 4F*) and upstream (28 nt RPFs) of the stalling site (compare bottom panels, *Figure 4C and 4F* and quantitation in 4H). These data were normalized to overall mRNA levels and reveal more than a twofold increase in ribosome occupancy when *SLH1* is deleted. These data are consistent with the proposed model that Slh1 prevents the accumulation of colliding ribosomes (*Matsuo et al., 2017*) thereby limiting the availability of Cue2 substrates.

## Genome-wide exploration of endogenous mRNA substrates of Cue2

One possible endogenous target class for Cue2 is the set of prematurely polyadenylated mRNAs (where polyadenylation happens within the ORF) that are commonly generated by aberrant RNA processing events in the nucleus. We looked for such candidates in our data sets by first identifying genes that exhibit evidence of untemplated A's within the monosome footprint population (i.e. ribosomes that are actively translating prematurely polyadenylated mRNAs) (*Figure 6A*, pink dots). These candidate genes nicely correlated with those previously identified from 3' end sequencing approaches (*Ozsolak et al., 2010*; *Pelechano et al., 2013*) (e.g. see data for two specific mRNAs *RNA14* and *YAP1* in *Figure 6—figure supplement 1A*) and from ribosome profiling approaches exploring Dom34 function (*Guydosh and Green, 2017*). We next identify Cue2-dependent 16 nt RPFs in a *ski2Δ dom34Δ* strain (*Figure 6A*, orange dots) as well as in a *ski2Δ dom34Δ slh1Δ* background (*Figure 6—figure supplement 1B*, orange dots) and see that they are substantially enriched in the prematurely polyadenylated mRNAs (*Figure 6A* and *Figure 6—figure supplement 1B*, red dots). We note that in the *ski2Δ dom34Δ* background a good fraction (21/55) of the prematurely polyadenylated genes are substantially reduced in 16 nt RPFs upon *CUE2* deletion (*Figure 6B*). Moreover, we show that for well-defined mRNA targets *RNA14* and *YAP1*, the Cue2-dependent 16 nt RPFs are positioned as anticipated directly upstream of the previously characterized sites of premature polyadenylation (*Figure 6C* and *Figure 6—figure supplement 1C*) (*Guydosh and Green, 2017*; *Pelechano et al., 2013*). These data support a role for Cue2-mediated endonucleolytic cleavage on such problematic transcripts. Although the SKI complex and the exosome decay NSD substrates (*Frischmeyer et al., 2002*; *van Hoof et al., 2002*), Cue2-cleavage may be one mechanism for cells to initiate this decay.

## Discussion

Cells have evolved a complex set of mechanisms to recognize and resolve ribosomes stalled on problematic mRNAs and to target the mRNAs for decay. The data presented here identify and characterize Cue2 as the endonuclease critical for NGD in yeast. We demonstrate using a wide range of in vivo and in vitro approaches that Cue2 is necessary and sufficient for cleavage of mRNAs loaded with stalled, colliding ribosomes. Structural homology modeling with the C-terminal domain of IF3 positions the catalytic SMR domain of Cue2 in the A site of the small ribosome subunit (*Figure 2C and 2D*; *Figure 2—figure supplement 1D to 1F*). And, according to this model, the conserved amino acids in Cue2 that closely approach mRNA in the A site (*Figure 2D*; *Figure 2—figure supplement 1B and 1C*) are indeed critical for efficient endonuclease cleavage activity (*Figure 2E and 2H*). Although mutating any one of the residues D348, H350, or R402, did not completely abolish Cue2 activity, mutating a combination of these residues or removing the entire SMR domain from the protein had a very strong effect (*Figure 2E and 2H*). Other studies have argued that the NGD endonuclease leaves a cyclic 3' phosphate and a 5' hydroxyl, products that are consistent with Cue2 being a metal independent endonuclease (*Navickas et al., 2018*). As metal independent RNases typically depend on mechanisms where RNA is contorted to act upon itself, it is often difficult to identify a

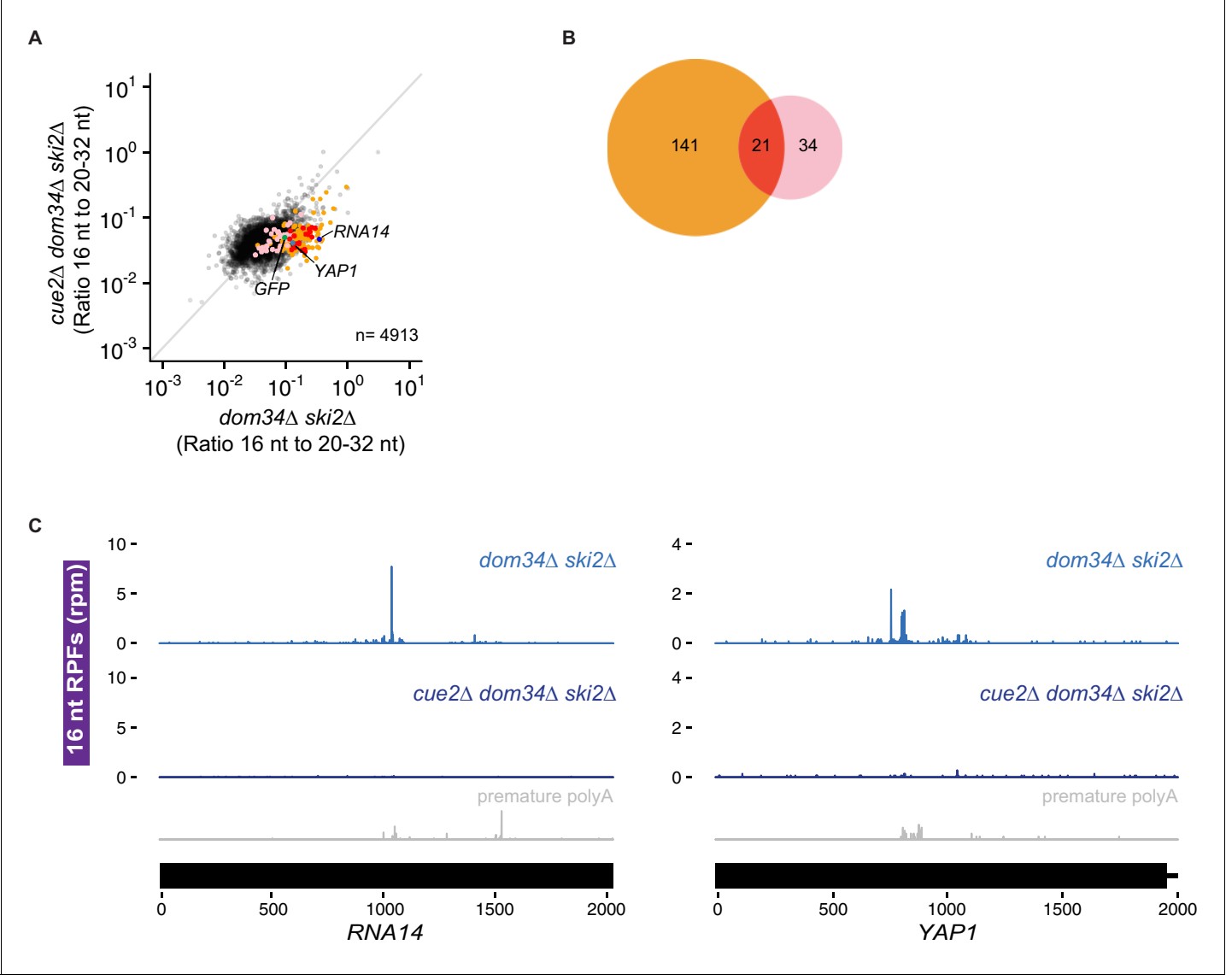

**Figure 6.** Cue2 targets prematurely polyadenylated mRNAs genome-wide for NGD. (**A**) Ratio of 16 nt over 20–32 nt RPFs plotted in *dom34Δ ski2Δ* and *cue2Δ dom34Δ ski2Δ* strains. Genes in orange correspond to those with reproducibly decreased 16 nt RPFs upon *CUE2* deletion. Pink dots indicate genes where premature polyadenylation was identified empirically from sequencing data. Red dots indicate the overlap between the pink and orange dots. NGD-CGA reporter, *YAP1* and *RNA14* are labeled and shown in green, blue and purple, respectively. (**B**) Overlap (red) between annotated prematurely polyadenylated genes (pink) and genes on which *CUE2* deletion had a substantial effect on 16 nt RPFs (orange). (**C**) Gene model examples of 16 nt RPFs mapped to genes, *RNA14* (left) and *YAP1* (right), in *dom34Δ ski2Δ* and *cue2Δ dom34Δ ski2Δ* strains with known premature polyadenylation sites indicated (gray traces; *Pelechano et al., 2013*). See *Figure 6—figure supplement 1*.

DOI: https://doi.org/10.7554/eLife.49117.013

The following figure supplement is available for figure 6:

**Figure supplement 1.** Cue2-dependent 16 nt RPFs on prematurely polyadenylated mRNAs.

DOI: https://doi.org/10.7554/eLife.49117.014

single essential catalytic residue. Interestingly, N4BP2, the mammalian homolog of Cue2, contains a PNK domain; this kinase activity could be required for the further degradation of the mRNA through Xrn1, which depends on 5' phosphate groups.

Strong support for the modeling of Cue2 in the A site came from the high-resolution ribosome profiling data establishing that Cue2 cleaves mRNAs precisely in the A site of the 'colliding' ribosome (*Figure 4*). We additionally find by profiling that the lead ribosome is stalled in an unrotated state (21 nt RPFs), unable to effectively accommodate the ICG-tRNA$^{Arg}$ that should normally decode

the CGA codon in the A site in concordance with previous studies on problematic codon pairs (*Wu et al., 2019b*). We also find that the lagging ribosomes are found in a pre-translocation or rotated state (28 nt RPFs). These different conformational states of the colliding ribosomes identified by ribosome profiling correlate with those defined in recent cryo-EM structures of colliding ribosomes (*Ikeuchi et al., 2019*; *Juszkiewicz et al., 2018*). We note that our data and existing structures suggest that Cue2 requires partial displacement of the peptidyl-tRNA substrate positioned in the A site of the colliding ribosome in order to access the mRNA. Recent studies identify residues in Rps3 at the mRNA entry tunnel that are required for NGD cleavage consistent with the idea that Cue2 binds within the ribosome mRNA channel to promote cleavage (*Simms et al., 2018*).

Another structural feature of Cue2 is that it possesses two highly conserved CUE domains at the N-terminus and up to two putative ubiquitin-binding domains prior to the proposed catalytic SMR domain. Endonucleolytic cleavage of problematic mRNAs was previously shown to depend on the function of the E3 ligase Hel2 which ubiquitinates several small subunit ribosome proteins (*Ikeuchi et al., 2019*) and our ribosome profiling data provides strong support for this model (*Figure 4—figure supplement 2*). We suspect that the ubiquitin-binding domains of Cue2 might recognize Hel2-ubiquitinated sites on the stacked ribosomes thus allowing recruitment of the SMR domain into the A site of the colliding ribosome. We can imagine that recognition of ubiquitin chains by Cue2 might involve multiple ubiquitinated sites on a single collided ribosome or those found on neighboring ribosomes. These conserved and putative ubiquitin-binding domains might also function within Cue2 itself to inhibit promiscuous activity of the endonuclease in the absence of an appropriate target, a possibility that prompted us to perform in vitro cleavage experiments with the isolated SMR domain of Cue2.

In addition to the identification and characterization of the key endonuclease involved in NGD, our results help to clarify the relative contributions of decay pathways for problematic mRNAs. We show that canonical exonucleolytic processing of problematic mRNAs is the dominant pathway on our NGD-CGA reporter, with the strongest contributions from Xrn1 rather than the exosome (*Figure 3*). Importantly, we find that the Cue2-mediated endonucleolytic pathway is activated in genetic backgrounds lacking the helicase Slh1 (a member of the previously identified RQT complex) (*Figure 2F–2G*). These ideas are brought together in the model in *Figure 7* that outlines how multiple decay pathways converge to bring about the decay of problematic mRNAs. We anticipate that mRNAs with different problematic features might differentially depend on these pathways as suggested by previous studies (*Tsuboi et al., 2012*). Importantly, however, our identification of the endonuclease for NGD will allow subsequent studies to better disentangle the distinct contributions from multiple decay machineries in the cell. A challenge moving forward will be to identify how problematic mRNAs with colliding ribosomes signal to decapping factors and Xrn1 to initiate degradation (indicated with question marks in the model in *Figure 7*).

Our genetic insights also lead us to speculate that under conditions where cells are overwhelmed by colliding ribosome complexes, for example under stressful RNA-damaging conditions (*Simms et al., 2014*), that the RQT machinery might become limiting and that there will be an increased role for Cue2-mediated endonucleolytic processing. Previous genetic screens reveal increased sensitivity of *CUE2* deletion strains to ribosome-targeted compounds (*Alamgir et al., 2010*; *Mircus et al., 2015*). More generally, the strong conservation of Cue2 throughout eukaryotes argues that this endonucleolytic processing pathway plays a fundamental role in biology.

Previous studies provide strong evidence that the activity of the RQT complex is critical in targeting nascent peptides on problematic mRNAs for degradation via the RQC (*Matsuo et al., 2017*; *Sitron et al., 2017*) through ribosome rescue events that generate a dissociated 60S-peptidyl-tRNA complex. Our ribosome profiling results indicate that ribosome occupancies on problematic mRNAs increase on deletion of *SLH1* (*Figure 4C, 4F and 4H*) consistent with the possibility that Slh1 functions directly or indirectly to dissociate ribosomes. The increased Cue2-dependent cleavage (*Figure 2F–2G* and *Figure 4B and 4E*) that we observe in *slh1Δ* background suggests that an increase in the density of collided ribosomes makes an ideal Cue2 substrate.

Taken together, we suggest that Slh1 and Dom34 may both target collided ribosomes for the RQC, although their specificities may be somewhat distinct. For example, Slh1 may use its helicase function to target ribosomes on intact mRNAs with an accessible 3' end, while Dom34 targets those ribosomes on truncated mRNAs that are inaccessible to Slh1 activity. In this view, a critical role of Cue2-mediated endonucleolytic cleavage is to provide another pathway for ribosome dissociation

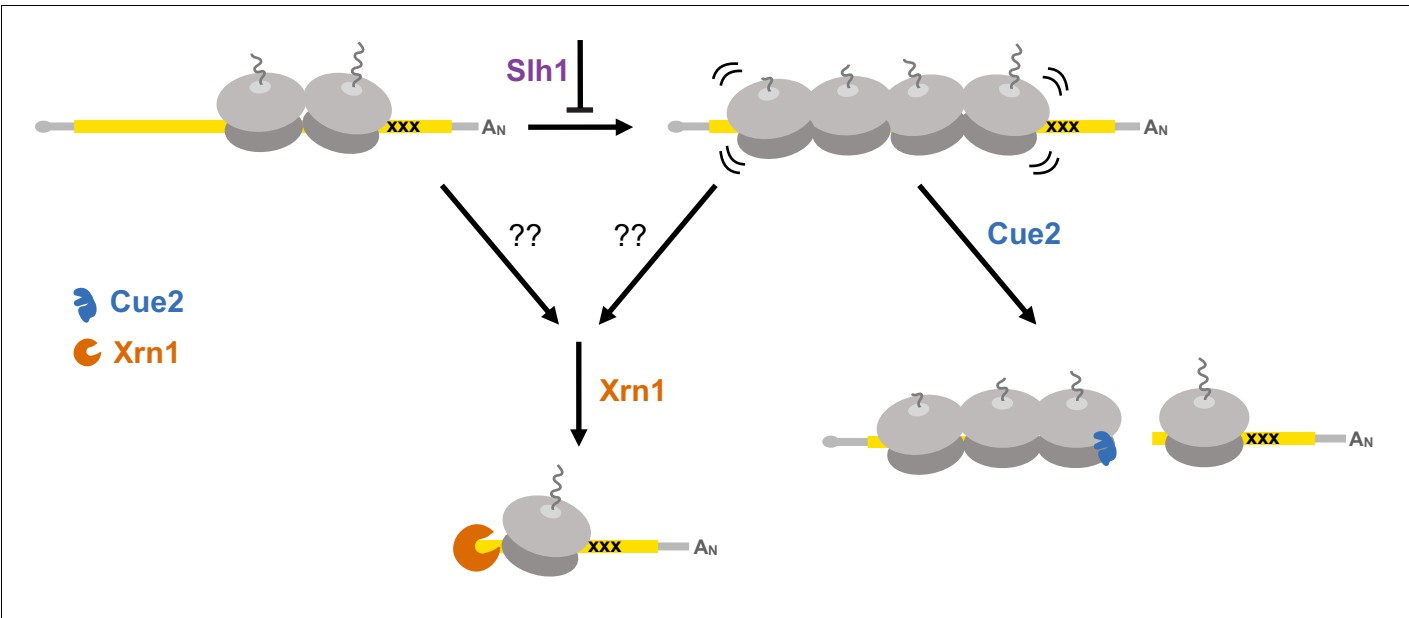

**Figure 7.** Multiple converging pathways for mRNA decay on NGD substrates. Ribosome stalling triggers decay either by the exonuclease Xrn1 or by the endonuclease Cue2. The relative contributions of these pathways are modulated by the activity of Slh1 (member of the RQT complex). Ribosomes are gray; open reading frame is yellow; 'xxx' indicates a stalling sequence in the mRNA.
DOI: https://doi.org/10.7554/eLife.49117.015

that can target nascent peptides for RQC. Previous studies provide strong support for the idea that Dom34 preferentially rescues ribosomes that are trapped on truncated mRNAs (*Guydosh and Green, 2014*; *Shoemaker et al., 2010*) and in vitro studies have connected this rescue activity of Dom34 to downstream RQC functions (*Shao et al., 2013*). These two distinct rescue pathways, one on intact mRNA by the RQT complex (and Slh1) and the other on endonucleolytically cleaved mRNA by Dom34, might provide the cell with redundant means to deal with toxic mRNP intermediates.

Proteotoxic stress is a critical problem for all cells and elaborate quality control systems have evolved to minimize its effects as made clear by the exquisite sensitivity of neurons to defects in these pathways (*Bengtson and Joazeiro, 2010*; *Ishimura et al., 2014*). The synergistic contributions of exonucleases and endonucleases that we delineate here for targeting problematic mRNAs for decay are critical for ensuring that proteotoxic stress is minimized. Future studies will delineate the biological targets and conditions in which these systems function.

## Materials and methods

**Key resources table**

| Reagent type (species) or resource | Designation | Source or reference | Identifiers | Additional information |
|---|---|---|---|---|
| Antibody | GFP primary antibody – mouse monoclonal | Takara | 632381 | (1:5000) |
| Antibody | FLAG primary antibody – mouse monoclonal | Sigma | F3165 | (1:5000) |
| Antibody | PGK1 primary antibody – mouse monoclonal | Invitrogen | 22C5D8 | (1:5000) |
| Antibody | HA primary antibody – rat monoclonal | Roche | 11867423001 | (1:5000) |
| Antibody | eEF2 primary antibody – rabbit polyclonal | Kerafast | ED7002 | (1:10000) |

*Continued on next page*

*Continued*

| Reagent type (species) or resource | Designation | Source or reference | Identifiers | Additional information |
|---|---|---|---|---|
| Software, algorithm | Rstudio | http://www.rstudio.com/ | | |
| Software, algorithm | Graphpad prism | www.graphpad.com | | |
| Software, algorithm | skewer | *Jiang et al., 2014* | Version 0.2.2 | |
| Software, algorithm | seqtk | https://github.com/lh3/seqtk | | |
| Software, algorithm | STAR | *Dobin et al., 2013* | STAR_2.5.3a_modified | |
| Software, algorithm | ImageQuant TL | GE Healthcare Life Sciences | Version 8.1 | |
| Software, algorithm | Spotfinder | *Saeed et al., 2003* | | |
| Software, algorithm | SGAtools | http://sgatools.ccbr.utoronto.ca/ | | |
| Software, algorithm | Custom software | https://github.com/greenlabjhmi/Cue2eLife | yeast_KD2.gff | Find information on use in Materials and methods section 'Analysis of ribosome profiling data' |
| Software, algorithm | Custom software | https://github.com/greenlabjhmi/Cue2eLife | StarAlignment_KD2.py | Find information on use in Materials and methods section 'Analysis of ribosome profiling data' |
| Software, algorithm | Custom software | https://github.com/greenlabjhmi/Cue2eLife | Screen_data_analysis.R | Find information on use in Materials and methods section 'Screen data analysis' |
| Strains | All strains | *Supplementary file 1* | | |
| Recombinant DNA reagent | All plasmids | *Supplementary file 1* | | |
| Sequence based reagent | All primers and oligos | *Supplementary file 1* | | |
| Peptide, recombinant protein | Cue2-SMR, and Cue2-SMR-R402A | This study | | Find information in Materials and methods section – 'Cue2 *E. coli* expression plasmids' and 'Cue2-SMR prep' |

## Plasmid construction
### GFP reporters

The OPT reporter plasmid, or pKD065, was cloned in pECB1806 (*Gamble et al., 2016*) as follows. Briefly, the *GAL1* promoter, a *GFP-2A-FLAG*, a partially non-optimal *HIS3* and the *ADH1* terminator were introduced in *Spel/Sphl*- digested pECB1806 using Gibson assembly leading to pKD064. The *GAL1* promoter was amplified by PCR using primers KD235 and KD236 and the *ADH1* terminator was PCR amplified using KD239 and KD240 with pECB1806 as template (*Supplementary file 1*). All PCR products were cleaned using Zymo Research DNA Clean and Concentrator Kit. The *GFP-2A-FLAG* and a partially non-optimal *HIS3* sequence fragment was gene-synthesized by Integrated DNA Technologies (IDT). The partially non-optimal *FLAG-HIS3* in pKD064 was then replaced by a fully

optimal sequence of *FLAG-HIS3* from pJC797 (*Radhakrishnan et al., 2016*) to make pKD065 (*Supplementary file 1*).

The NGD-AAA and NGD-CGA reporter plasmids (pKD079 and pKD080, respectively) were cloned using pKD065. In brief, a (AAA)x12 or (CGA)x12 codon sequence was inserted 90 codons into the *HIS3* gene of pKD065. The PCR product from (AAA)12 primers (KD281 and KD280) or (CGA)12 primers (KD283 and KD280) with pKD065 template was combined with the PCR product from primers KD287 and KD276 with pKD065 template and *Sall*/*Sph*1 digested pKD065 and these products were Gibson assembled to produce pKD079 and pKD080 (*Supplementary file 1*).

### HA-CUE2 overexpressing plasmid
The CUE2 overexpressing FLEX plasmid (pKD100) was rescued from the yeast FLEX library (*Hu et al., 2007*; *Kainth et al., 2009*). A gene block (gKD002) of truncated CUE2 (NoSMR) was inserted into *BamHI*/*Sph*I digested pKD100, to make pKD105. 5' 3xHA-tagged *CUE2*, and *NoSMR* were PCR amplified using KD414 and KD416 and Gibson assembled with *BamHI*/*Sph*I digested pKD100 to make pKD120 and pKD125, respectively (*Supplementary file 1*).

### Site-directed mutagenesis
Using the standard protocol for the QuikChange Lightning Multi Site-Directed Mutagenesis Kit from Agilent Technologies, the indicated mutations in the *HA-CUE2* construct (pKD120) were made to make pKD127, pKD129, pKD131, pKD133, and pKD145 (*Supplementary file 1*).

### CRISPR plasmids
BpII digested plasmid pJH2972 (Anand, Memisoglu, and Haber, n.d.) was used in a Gibson reaction with primers KD503, KD505, and KD509 to make plasmids pKD163, pKD165, and pKD169 respectively.

### Cue2 *E. coli* expression plasmids
PCR products from KD457 and KD338 from plasmids pKD120 and pKD129 were Gibson cloned into BamHI and XhoI digested pSMT3, to make pKD097 and pKD098, respectively.

## Yeast strains and growth conditions
Yeast strains used in this study are described in *Supplementary file 1* and are all derivatives of BY4741 unless specified otherwise. Yeast strain construction was performed using standard lithium acetate transformations. Reporters strains were constructed by integrating the various GFP-2A-FLAG-HIS3 cassettes, from *Stu*I digested pKD065, pKD079, and pKD080, into the *ADE2* locus of BY4741 (*Supplementary file 1*). For SGA experiments, query strains overexpression screens were constructed by introducing the GFP-2A-FLAG-HIS3 cassettes from *Stu*I digested pKD065, pKD079, or pKD080 at the *ADE2* locus in BY4741 (*Supplementary file 1*).

Deletion strains were constructed by inserting resistance cassettes from *Longtine et al. (1998)* into the designated loci and genotypes are listed in *Supplementary file 1*. Note: two different deletion strains were used for *XRN1* deletions in this study. See *Supplementary file 1* for genotypes.

HA tag insertions, point mutations, and the SMR deletion were made in yKD143 as described in *Anand et al. (2017)*, using plasmids pKD163, pKD165, and pKD169 and homology directed repair templates.

Recipes for media used in this study are listed in *Supplementary file 1*.

For gal-induced growths, overnight cultures were grown in YPAGR media, or, for strains with plasmids, overnight cultures were grown in SC/A/GR/-Ura. Overnights were diluted in the same media to an OD of 0.1 and harvested at an OD of 0.4–0.5.

## Flow cytometry
### Data collection
100 µl of log-phase cells were pelleted and washed once with 1 x PBS. Cells were then resuspended in 500 µl of 1 x PBS and 5000 cells were analyzed with a Millipore guava easyCyte flow cytometer for GFP and RFP detection using 488 nm and 532 nm excitation lasers, respectively.

## Data analysis

Cells were gated based on size, and random outliers were cut off from graphs for visual purposes (plot values were not changed upon removal of outliers). For all flow cytometry data, violin plots show the density of cells for $\log_2$(GFP/RFP) values for individual cells. Flow cytometry was done in triplicate, with each group of cells taken from individual growths. For triplicate plots, the average of each individual flow cytometry sample was taken and plotted. Machine settings remained constant between samples.

For statistical analysis of flow cytometry triplicates, a standard t-test was run in R and p-values are reported.

## Northern blots

### RNA isolation

25 ml of log-phase cells were pelleted and supernatant was poured off. Cells were resuspended in residual media, pelleted again, and flash frozen in liquid nitrogen. Cell pellet was resuspended in buffer with 8.4 mM EDTA, and 60 mM NaOAc. 20% SDS was added to a final concentration of 1.5%. Cell solution was warmed at 65°C for 2 min and added to acid phenol at 65°C. Phenol/cell solution was shaken at 1100 rpm at 65°C for 10–20 min with intermittent vortexing. Samples were put on ice for 5 min then spun at 16 krpm. The aqueous layer was removed and added to an equal volume of phenol, and samples were vortexed again. Samples were spun at 16 krpm again, the aqueous layer was removed and added to an equal volume of chloroform, and samples were vortexed. Lastly, samples were spun again at 16 krpm, the aqueous layer was removed and precipitated in NaOAc and isopropanol. The RNA pellet was resuspended in 10 mM Tris-HCl, pH 8.0.

### Northern blot

Between 5 and 10 µg of RNA were loaded into a 1.2% agarose, formaldehyde denaturing gel and run for 2–2.5 hr at 125 volts (for any given gel, the same amount of RNA was loaded for each sample, but this varied from gel to gel). The RNA was vacuum transferred to a nitrocellulose (N + Hbond, Amersham) membrane in 10 x SSC buffer. The RNA was then UV crosslinked to the membrane and placed in pre-hybridization buffer and rotated at 42°C. The indicated PNK-end labeled probe was added to the pre-hybridization buffer at 42°C after 30 min (*Supplementary file 1*). The membrane was probed overnight, rotating at 42°C. The membrane was washed 3 times in 2 x SSC, 0.1% SDS for 20 min at 30°C, then exposed to a phosphoscreen. The phosphoscreen was scanned using a Typhoon FLA 9500.

### End-labeled DNA probe

This indicated DNA-oligo listed in *Supplementary file 1* was end labeled using gamma-ATP and the standard T4 Polynucleotide Kinase radioactive labeling protocol from NEB. The labeled oligo was purified using GE Healthcare illustra ProbeQuant G-50 Micro Columns.

## Western blots

### Protein isolation

2 OD units of log-phase cells were pelleted and supernatant was poured off. Cells were resuspended in residual media, pelleted again, and flash frozen in liquid nitrogen. Pellets were resuspended in 200 µl lysis buffer containing 20 mM Tris-HCl, pH 8.0, 140 mM KCl, 5 mM MgCl2, 1% triton, 1 mM DTT, Roche cOmplete protease inhibitor tablet, and PMSF, pepstatin, and leupeptin protease inhibitors. The volume of cell solution was approximately doubled using acid-washed glass beads and cells were mechanically lysed using bead beater 1 min on, 1 min off, for three cycles. 6x SDS loading dye was added to the lysis solution at 2x and samples were boiled for 5 min.

### Western blot

Equal volumes of lysate were loaded for each sample onto 4–12% Criterion XT Bis-Tris protein gels in 1x XT MES buffer. Protein was transferred to PVDF membrane via turbo blot. Membranes were then placed in 2.5% milk, 1x TBST blocking solution for 1 hr. Primary antibody was used in 1x TBST at either 1:50,000 for anti-eEF2 (Kerafast, rabbit), or 1:5000 for anti-PGK1 (Invitrogen, mouse), anti-

HA (Roche, rat), anti-FLAG (Sigma, mouse), and anti-GFP (Takara, mouse) and incubated on a rotator overnight at 4°C. Membranes were washed in 1x TBST three times, 10 min each. The corresponding HRP-conjugated secondary antibody was added to the membrane at 1:5000 in 1x TBST and incubated for 1–2 hr. Membranes were washed three times for 10 min each. Pico solution (details) was added to membranes for approximately 3 min and then membranes were scanned using a G:BOX Chemi XX6 (Syngene) with varying exposure times.

## Reporter-SGA screens
### SGA procedure
SGA screens were performed using a Biomatrix Robot (S and P Robotics Inc) with a few modifications (*Tong and Boone, 2006*). Briefly, yKD131, yKD132, yKD133 query strains (*Supplementary file 1*) were crossed individually with the FLEX collection (*Douglas et al., 2012*; *Hu et al., 2007*). The FLEX library was arrayed in a 1536-format containing four colonies for each overexpression strain. Mating steps were performed on standard SGA media (*Tong and Boone, 2006*).

For the overexpression screen, diploid strains were selected on OEDIP media listed in *Supplementary file 1*. Sporulation and haploid double mutant selection steps were not performed in this screen.

To induce reporter expression and FLEX gene overexpression, cells were pinned onto the same medium (OEDIP), with the exception that glucose was replaced by raffinose and galactose at a final concentration of 2% (OEDIPGR media listed in *Supplementary file 1*). Cells were grown for 40–46 hr for the overexpression screen before scanning on a Typhoon FLA9500 (GE Healthcare) fluorescence scanner equipped with 488 nm and 532 nm excitation lasers and 520/40 and 610/30 emission filters. Plates were also photographed using a robotic system developed by S and P Robotics Inc in order to determine colony size.

### Screen data analysis
GFP and RFP fluorescent intensity data was collected using the microarray software, TIGR Spotfinder (*Saeed et al., 2003*). Colony size data was aquired using SGATools (*Wagih et al., 2013*) (http://sga-tools.ccbr.utoronto.ca/). Subsequent data analysis was performed as previously described (*Hendry et al., 2015*; *Kainth et al., 2009*). In brief, border strains and size outliers (<1500 or >6000 pixels) were eliminated, and median GFP and RFP values were taken for the remaining strains. $\log_2$(mean GFP/mean RFP) values were then calculated and LOESS normalized for each plate. Finally, Z-scores for each individual plate were calculated based on the LOESS normalized $\log_2$(mean GFP/mean RFP). Strains for the AAA or CGA reporters with Z-scores greater than 2.5 or less than −2.5 were considered as hits if their Z-score in the OPT reporter was unaffected (i.e had a Z-score between −2.5 and 2.5). For subsequent experiments, hits were reconstructed in the background strain yKD133 as described above and all experiments were performed with the reconstructed strains.

### Venn diagrams
Overexpression screen strains with a Z-score greater than 2.5 or less than −2.5 were considered candidate genes and analyzed for overlap using BioVenn (http://www.biovenn.nl/) to produce the diagrams.

## Cue2-SMR prep
The SMR and the R402A-SMR constructs, pKD097 and pKD098, respectively, were expressed in RIPL BL21 *E. coli* strain using Kan resistance and inducing at OD = 0.4, at 18°C with IPTG at 0.5 mM overnight. Cells were harvested by centrifugation and then flash frozen in liquid nitrogen. Pellets were then resuspended in protein prep lysis buffer [25 mM Tris-Cl pH 7.5, 500 mM KCl, 1 mM MgCl2, 5 mM bMe, 10% glycerol]+PMSF, leupeptin, pepstatin, and Roche cOmplete EDTA-free protease inhibitors.

Cells were lysed using the French press, three times, 1100 pressure units. Lysis solution was clarified at 20,000xg for 30 min. Lysate was filtered through a 0.2 um filter and run over HiTrap 5 ml Ni-NTA column. Column was washed for 5 CV in lysis buffer with high salt (1 M KCl) and 20 mM imidazole. Sample was batch eluted using Ni elution buffer [25 mM Tris-Cl pH 7.5, 500 mM KCl, 1 mM

MgCl2, 5 mM bMe, 10% glycerol, 500 mM imidazole]. Sample was diluted 10 fold into S column buffer [25 mM Tris-Cl pH 7.5, 200 mM KCl, 5 mM MgCl2, 5 mM bMe, 10% glycerol] and then run on a Resource S column. Sample was washed in S column buffer for 5 CV and then gradient eluted off of S column in S column buffer with 1 M salt. 6xHIS-Sumo tag was cleaved overnight using SUMO protease. And sample was run on an orthogonal Ni column to remove tag. Flow through was collected and concentrated to 40 uM SMR, and 20 uM SMR-R402A.

## Isolation of nuclease-resistant trisomes

Grow 1 L of yKD307 in YPAGR to an OD 0.4. For cycloheximide treated cells, added 1 mg/L cycloheximide, grew for 30 min and filter harvested. For untreated cells, immediately filter harvested. Cell pellets were ground with 1 mL lysis buffer [10 mM potassium phosphate (pH 6.1), 140 mM KCl, 5 mM MgCl$_2$, 1% Triton X-100, 0.1 mg/mL cycloheximide, 1 mM DTT, Roche cOmplete EDTA-free protease inhibitor, leupeptin, PMSF, pepstatin] in a Spex 6870 freezer mill. Cell lysates were clarified by centrifugation. CaCl2 was added to 200 OD units of clarified lysates to a final concentration of 2.5 mM and the indicated clarified lysates were treated with 20 µl of NEB MNase at 2e̊six gel units/ml for 30 min. 4 mM EGTA was added to quench the digestion. Samples were layered on a 15–45% sucrose gradient [10 mM potassium phosphate (pH 6.1), 140 mM KCl, 5 mM MgCl$_2$] and spun in SW28 rotor at max speed for 4 hr. Nuclease resistant trisome peak was isolated and diluted two fold in lysis buffer.

## Cleavage of lysates with SMR and analysis

Protein prep buffer, the purified SMR domain of Cue2, and the R402A mutant of the SMR domain of Cue2 were added to separate aliquots of isolated nuclease-resistant trisomes, such that the final concentration of enzyme (for samples with enzyme)=3 µM. Samples were left at room temperature for 2 hr. Next, samples were run on a 15–45% sucrose gradient [10 mM potassium phosphate (pH 6.1), 140 mM KCl, 5 mM MgCl$_2$] and the A260 trace is reported. RPFs were extracted from pooled fractions (mono-, di-, and trisomes).

## Sequence alignment

Information from NCBI conserved domain database (*Marchler-Bauer et al., 2011*), conserved domain architecture retrieval tool (*Geer et al., 2002*), available structures, and Phyre-based homology modeling (*Kelley et al., 2015*) was used to define domain boundaries. Structure-based multiple sequence alignments were carried out using Expresso (*Armougom et al., 2006*), and illustrated with ESPript (*Robert and Gouet, 2014*). Protein names are indicated, followed by the name of organism, and residues used for alignment. YEAST, *Saccharomyces cerevisiae*; CANGA, *Candida glabrata*; ARATH, *Arabidopsis thaliana*; HUMAN, *Homo sapiens*; MACMU, *Macaca mulatta*; BOVIN, *Bos taurus*; THETH, *Thermus thermophilus*; BACSU, *Bacillus subtilis*; ECOLI, *Escherichia coli*.

## Homology modeling

Homology model of the Cue2 SMR domain templated on human N4BP2 SMR domain (PDB: 2VKC [*Diercks et al., 2008*]) was generated using Phyre 2.0 (*Kelley et al., 2015*) and SWISS-MODEL (*Waterhouse et al., 2018*). After identifying N4BP2 as a Cue2 homolog, a heuristic DALI search was performed using the N4BP2 SMR domain (PDB IDs: 2D9I and 2VKC) as query against structures in the Protein Data Bank (PDB). This exercise, along with templates identified using Phyre 2.0 (*Kelley et al., 2015*) and SWISS-MODEL (*Waterhouse et al., 2018*), enabled us to identify structural conservation between the SMR domain of Cue2 and the C-terminal domain (CTD) of translational initiation factor 3 (IF3). This was further validated using structure-based sequence alignments as shown in *Figure 2—figure supplement 1B*. Structural comparisons and fittings were performed using UCSF Chimera (*Goddard et al., 2007*; *Pettersen et al., 2004*). Superimposition of the CTD of IF3 (PDB: 1TIG [*Biou et al., 1995*]) and the Cue2-SMR homology model revealed structural conservation between the two domains (*Figure 2—figure supplement 1C*). This enabled superimposition of the Cue2 SMR in the context of full-length *Thermus thermophilus* IF3 (PDB: 5LMQ [*Hussain et al., 2016*]) (see *Figure 2C*). This fitting enabled overlay of the Cue2-SMR at the A/P-site in the context of IF3 and tRNA$^{fMet}$-bound 30S pre-initiation complex (PDB: 5LMQ, State 2A [*Hussain et al., 2016*]) (see *Figure 2D* and *Figure 2—figure supplement 1, D–E*). The mRNA and the 30S ribosomal

subunit are represented as surfaces to minimize over interpretation (*Figure 2D* and *Figure 2—figure supplement 1, D–E*).

## Yeast growth conditions for ribosome profiling

Overnight seed cultures were grown in YPAD at 30°C. To induce NGD-CGA reporter expression, cells were collected by centrifugation, washed, and resuspended in YPAGR. Cells were then harvested at OD ~0.5 by fast filtration and flash frozen in liquid nitrogen.

## Preparation of libraries for yeast ribosome footprints

Cell pellets were ground with 1 mL footprint lysis buffer [20 mM Tris-Cl (pH8.0), 140 mM KCl, 1.5 mM MgCl2, 1% Triton X-100, 0.1 mg/mL cycloheximide, 0.1 mg/mL tigecycline] in a Spex 6870 freezer mill. Lysed cell pellets were diluted to 15 mL in footprint lysis buffer and clarified by centrifugation. The supernatant was layered on a sucrose cushion [20 mM Tris-Cl (pH8.0), 150 mM KCl, 5 mM MgCl2, 0.5 mM DTT, 1M sucrose]. Polysomes were pelleted by centrifugation at 60,000 rpm for 106 min in a Type 70Ti rotor (Beckman Coulter). Ribosome pellets were gently resuspended in 800 μL footprint lysis buffer. 350 μg of isolated polysomes were treated with 500 units of RNaseI (Ambion) for 1 hr at 25°C. Monosomes or disomes were isolated by sucrose gradients (10–50%). RNA was extracted by hot acid phenol and then size-selected from 15% denaturing PAGE gels, cutting between 15–34 nt for monosome footprints, 40–80 nt for disome footprints, and 15–90 nt for in vitro Cue2 cleavage assays. Library construction was carried out as described (*Wu et al., 2019b*). Libraries were sequenced on a HiSeq2500 machine at facilities at the Johns Hopkins Institute of Genetic Medicine.

## Analysis of ribosome profiling data

The R64-1-1 S288C reference genome assembly (SacCer3) from the Saccharomyces Genome Database Project was used for yeast genome alignment. For the rest of our libraries, 3' adapter (NNNNNNCACTCGGGCACCAAGGA) was trimmed, and four random nucleotides included in RT primer (RNNNAGATCGGAAGAGCGTCGTGTAGGGAAAGAGTGTAGATCTCGGTGGTCGC/iSP18/TTCAGACGTGTGCTCTTCCGATCTGTCCTTGGTGCCCGAGTG) were removed from the 5' end of reads with skewer (54). Trimmed reads were aligned to yeast ribosomal and non-coding RNA sequences using STAR (49) with '–outFilterMismatchNoverLmax 0.3'. Unmapped reads were then mapped to genome using the following options '–outFilterIntronMotifs RemoveNoncanonicalUnannotated –outFilterMultimapNmax 1 –outFilterMismatchNoverLmax 0.1'. All other analyses were performed using software custom written in Python 2.7 and R 3.3.1.

For monosome footprints, the offset of the A site from the 5' end of reads was calibrated using start codons of CDS (*Wu et al., 2019b*). For 28 nt RPFs, offsets (27:[16], 28:[16], 29:[17], 30:[17], 31:[17], 32:[17]) were used to infer the A sites of 27–32 nt reads. For 21 nt RPFs, offsets (20:[16], 21:[17], 22:[17]) were used. For 16 nt RPFs, 3' ends of 15–17 nt RPFs were used to infer cleavage sites. For 60 nt disome footprints, offsets (57:[47],58:[47],59:[47],60:[47],61:[47],62:[47]) were used to infer the A sites of lead ribosomes. For 54 nt disome footprints, offsets (51:[47],52:[47],53:[47],54:[47]) were used. For 46 nt disome footprints, 3'ends of 44–48 nt RPFs were used to infer cleavage sites. For in vitro cleavage assays, 3'ends of 60–65 nt were used to infer Cue2 SMR cleavage sites. Prematurely polyadenylated mRNAs were identified by monosome footprints (15–34 nt) with the criteria of at least three RPFs of more than one untemplated A's in the 3'end from both slh1Δ cue2Δ dom34Δ ski2Δ datasets. Cue2 target genes were identified using 15–17 nt RPFs that exhibited reproducible reduction upon Cue2 deletion (adjusted p<0.005, DESeq [*Anders and Huber, 2010*]).

## Accession number

Raw sequencing data were deposited in the GEO database under the accession number GSE129128.

## Data and materials availability

Raw sequencing data were deposited in the GEO database under the accession number GSE129128.

## Acknowledgements

We thank Miguel Pacheco for help with yeast experiments during the crunch. We thank Brendan P Cormack for helpful consultation on ideas and techniques throughout this study, Ryan McQuillen for help with figures, and members of the Greider lab, Carla Connelley, Rebecca Keener, and Calla Shubin, for assistance with techniques and careful interpretations of yeast genetic experiments. We thank Allen R Buskirk, Boris Zinshteyn, Daniel Goldman, Kamena K Kostova, Chirag Vasavda, and Jamie Wangen for careful reading of the manuscript and all Green lab members for helpful discussions throughout this study.

## Additional information

### Competing interests

Rachel Green: Reviewing editor, *eLife*. The other authors declare that no competing interests exist.

### Funding

| Funder | Grant reference number | Author |
|---|---|---|
| National Institutes of Health | R37GM059425 | Rachel Green |
| National Institutes of Health | 5T32GM007445-39 | Rachel Green |
| Canadian Institutes of Health Research | FDN-159913 | Grant W Brown |

The funders had no role in study design, data collection and interpretation, or the decision to submit the work for publication.

### Author contributions

Karole N D'Orazio, Conceptualization, Data curation, Formal analysis, Investigation, Methodology, Writing—original draft, Project administration, Writing—review and editing; Colin Chih-Chien Wu, Data curation, Formal analysis, Investigation, Writing—original draft, Writing—review and editing; Niladri Sinha, Investigation, Visualization, Writing—review and editing; Raphael Loll-Krippleber, Supervision, Investigation, Methodology, Project administration, Writing—review and editing; Grant W Brown, Resources, Supervision, Investigation, Methodology, Writing—review and editing; Rachel Green, Conceptualization, Resources, Supervision, Funding acquisition, Methodology, Writing—original draft, Writing—review and editing

### Author ORCIDs

Karole N D'Orazio ⓘ https://orcid.org/0000-0003-1954-4406
Colin Chih-Chien Wu ⓘ https://orcid.org/0000-0002-6013-0224
Niladri Sinha ⓘ https://orcid.org/0000-0002-9143-495X
Rachel Green ⓘ https://orcid.org/0000-0001-9337-2003

### Decision letter and Author response

Decision letter https://doi.org/10.7554/eLife.49117.021
Author response https://doi.org/10.7554/eLife.49117.022

## Additional files

### Supplementary files

• Supplementary file 1. Methods supplementary table.
DOI: https://doi.org/10.7554/eLife.49117.016

• Transparent reporting form
DOI: https://doi.org/10.7554/eLife.49117.017

## Data availability

Sequencing data have been deposited in NCBI Gene Expression Omnibus under accession code GSE129128.

The following dataset was generated:

| Author(s) | Year | Dataset title | Dataset URL | Database and Identifier |
|---|---|---|---|---|
| Wu CC, D'Orazio KN, Green R | 2019 | Cue2 and Slh1 define parallel pathways to rescue stalled ribosomes | http://www.ncbi.nlm.nih.gov/geo/query/acc.cgi?acc=GSE129128 | NCBI Gene Expression Omnibus, GSE129128 |

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
