## [Decision Letter]

Thank you for choosing to send your work entitled "The endonuclease Cue2 cleaves mRNAs at stalled ribosomes during No Go Decay" for consideration at *eLife*. Your submission, the previous reviews, and your responses have been evaluated by a Senior Editor (James Manley) and a Reviewing Editor (Nahum Sonenberg), and we would be happy to accept the present version for publication by *eLife*.

The authors have added important data showing in vitro activity of Cue2, including use of a mutant. They have also amended the paper to include missing information concerning the experimental repeats. Considering the novelty of the data and the great interest in this area of research, we recommend acceptance.

---

## [Author Response]

[Editors' note: we include below the reviews that the authors received from another journal, along with the authors’ responses.]

As this manuscript has previously been reviewed, we have included these reviews with the hope that this might expedite the process. While all three reviewers acknowledged the strong contribution of the study, each had general concerns about the absence of direct biochemical evidence for Cue2 function, and two had concerns about our incomplete definition of the Slh1-mediated ribosome clearing pathway that we included in our model.

Most critically, we include analysis of mutations in the endogenous Cue2 locus as well as in vitro biochemistry to further substantiate Cue2 endonuclease activity, as well as more carefully stating our conclusions regarding the molecular function of Slh1.

These studies were based on an initial genetic screen and used a very wide array of genetic and ribosome profiling approaches to yield critical insights on mechanistic aspects of these pathways. And, while these studies were performed in yeast (where the genetics and profiling are powerful), the protein factors that we characterize (Cue2, Slh1, Hel2, and others) are conserved in higher eukaryotes (and mammals). We anticipate that this work will have broad appeal for the community and importantly provides strong insights into the molecular complexities of these pathways.

Reviewer 1:

This manuscript addresses the mechanism by which stalled elongating ribosomes are released from yeast mRNAs. The main conclusions of the work are i) to argue that Cue2 is the endonuclease that cleaves NGD substrates, and ii) that stalled ribosomes can be released either by Cue2 dependent cleavage of the mRNAs, or in a parallel pathway by the action of the SLH1 helicase.The strengths of the work are i) the possible identification of Cue2 as the NGD endonuclease, and ii) evidence suggesting two parallel pathways for ribosome release. If these conclusions were well documented, the work could be appropriate for publication.The weaknesses of the work are i) that the evidence for Cue2 being the endonuclease is limited, ii) that while it is interesting to show there are two parallel mechanisms, the work would be more complete with an analysis of what mRNAs those pathways affect and their biological significance. Specific suggestions to this effect are below.Major Comments:1) I found the conclusion that Cue2 is the endonuclease for NGD to not be robust. To illustrate I review the evidence leading to this conclusion:i) OE of Cue2 protein reduces mRNA levels of full length reporter mRNAs, and cue2# reduces the levels of mRNA decay fragments, which together show Cue2 is involved in NGD but does not define mechanism.ii) That Cue2 has a domain that in other proteins can cleave DNA or RNA, which is suggestive but not conclusive.iii) That some cue2 mutations, which based on homology models to IF3 might be near the mRNA, alter the ratio of GFP to RFP, which is an indirect readout for mRNA levels. On the basis of this data, and since the 16mer RPF fragments (indicative of mRNA cleavage) are still produced to some level in the cue2# dom34# strain, I personally would be nervous about concluding that Cue2 is the endonuclease for NGD. I would suggest either: a) changing the conclusion to Cue2 is required for endonuclease cleavage in NGD, and might be the endonuclease, or b) If the main point of the manuscript is to identify the NGD endonuclease, then obtain more convincing data that Cue2 catalyzes endonuclease cleavage.Stronger data for Cue2 being the endonuclease could come from i) a better understanding of the active site of this type of domain and showing mutations expected to affect catalysis alter mRNA fragment production, ii) looking directly at NGD cleavage in cells by assaying fragment production for mutations thought to affect catalysis instead of looking at the indirect assay of GFP/RFP ratios, iii) examining if OE of CUE2 increases cleavage fragment production, and iv) examining RNase activity of Cue2 in vitro, and showing mutations that alter that activity alter NGD cleavage in vivo.I realize that it may be difficult (and possibly beyond the work of this manuscript) to demonstrate Cue2 endonuclease cleavage in vitro. Indeed, it may be that the endonuclease activity in NGD is actually more of a composite site (similar to MutS), with the active site made up of ribosomal components, Mg++, and parts of Cue2 or other proteins. Thus, my specific suggestion for this manuscript is:a) Make a more refined analysis of the Cue2 putative active site, perform a more extensive mutagenesis of Cue2, and then test the functions of those mutations on NGD cleavage by northern blots. This might allow a stronger argument that Cue2 functions as an endonuclease.b) At least try assaying RNase activity of the purified Cue2 protein (or domain). While a failure to demonstrate endonuclease activity would not rule out Cue2 is part of an endonuclease site, a demonstration of even weak endonuclease activity (that was affected by mutations in vitro that also affected NGD cleavage in cells), would provide convincing data that Cue2 is an endonuclease functioning in NGD.

Point 1: The reviewer makes excellent suggestions for how to improve the strength of our claims that Cue2 is the endonuclease and we have now included these experiments. In addition to making mutations in CUE2 and testing in the overexpression assay (as we did previously in Figure 2), we also incorporated these same mutations at the endogenous locus, with the incorporation of an HA tag to follow protein levels, and showed that these mutations resulted in the loss of generation of the cleavage fragment by northern analysis (new Figure 2H). We also include in vitro biochemistry with purified SMR domain from Cue2 (and the R402A mutation) and purified colliding ribosomes, and show by sucrose gradient sedimentation (Figure 5A) and by sequencing (Figure 5B) that purified protein generates cleavages of the NGD-CGA reporter that matches observed cleavage sites in vivo as defined by ribosome profiling.

2) The second major conclusion of this work is to argue that for parallel ribosome release pathways due to Slh1 (allowing ribosome release) and Cue2 (promoting cleavage and Dom34 dependent release). While the data for these parallel pathways is relatively convincing, no insight is provided as to the biological significance of two parallel pathways. Since the authors have generated genome wide RPF data for a variety of strains, they should be able to assess if there are specific mRNAs where Slh1 is promoting ribosome release (and ribosomes accumulate in the slh1# cells), and/or Cue2 is promoting cleavage events (where 16mer RPFs disappear in the cue2# cells). If one could identify that some mRNAs are affected by one or the other pathway it would allow one to describe the biological significance of these parallel pathways, and thereby make the discovery of parallel pathways have impact.

Point 2: This suggestion is an interesting one. It is possible that there is much overlap in cellular targets of Slh1 and Cue2 or that some are common and others are distinct. However, as we do not currently have an ideal metric for identifying Slh1 targets and as the manuscript is now more narrowly focused on Cue2 and its endonuclease activity, we are leaving this interesting question to more long-term studies.

3) I found Figure 3 and its accompanying text difficult to follow for three reasons. First, yeast strains being compared are sometimes split between two different figures (Figures 3 and S4) with different concentrations of CYH. Since these types of assays are quite sensitive to growth time, all relevant and compared strains should be shown on the same plates.Second, all pertinent strains need to be shown (including strains without ski2#) in order to conclude the nature of a genetic interaction. For example, the authors conclude that the dom34# ski2# strain is sensitive to CYH, and that is partially rescued by the cue2#. It looks to me that cue2# also rescues the CYH sensitivity of the ski2# alone. Does a cue2# by itself increase resistance to CYH? Does this suppression only occur in the ski2# background (which might inactivate roles of Ski7 in ribosome release)? [In another example, it seems the most interesting double mutant would be the slh1# cue2# strain. What happens in this strain?]Third, the interpretation of what these genetic interactions mean is possible, but interpreting mechanistic effects solely from genetic interactions about drug sensitivities is difficult.My recommendation here: a) Show all relevant strains on the same plates to allow the readers to be convinced by differences in growth side by side. b) Consider putting this experiment after the RPF footprinting and then interpret the basis of the drug interaction phenotypes on the basis of what happens in the relevant strains to ribosomes by footprinting. c) To make the experiments clear: Label strains as to what they are. I recommend including the ski2# in all genotypes in both Figure 3 and 4 to be up front about the experiment.

Point 3: We appreciate the reviewer’s concerns and the limitations of this experiment. While we believe our original data provided some thought provoking outcomes, we admit that it is hard to make direct molecular connections to this broad in vivo phenotype. In the revised manuscript, we have left out the survival plating assays as we do not yet know the precise conditions under which Cue2 is required for cell growth. We do add a line in the discussion about screening data from other laboratories which suggest that Cue2 is most critical when ribosomes are in distress.

Additional Minor Points:4) In Figure S2F, the authors show westerns of the mutant Cue2 proteins to demonstrate they are equally expressed. However, without some type of loading control, this is not very convincing.

Point 4: We have repeated this, along with replicates for the flow cytometry data throughout the manuscript, and now include a loading control (new Figure S2G and S2H). The expression levels of wild type and mutant variants of CUE2 were roughly equal.

5) In the text, the authors describe "Simultaneously mutating two of these conserved residues, H348 and D350, nearly restores GFP reporter signal.". From my interpretation of the figure, this is actually a triple mutant including a F to A change. The text should be clear about what the mutation is to allow the reader to understand the limitations of the presented experiment (what happens in the F349A mutant by itself?).

Point 5: Agreed – we have corrected this – it was unintentionally confusing.

6) Why isn’t the slh1# RPF shown in the main figure? It belongs there.

Point 6: We have included this additional panel in the revised manuscript (Figure 4E).

Reviewer 2:

In this study, the Green lab aims to use a combination of genetic screening, ribosome profiling, and more targeted reporter analysis to identify key contributors to mRNA decay downstream of ribosome stalling (no-go decay, or NGD). For around 15 years, it was thought that a key initial step in NGD is endonucleolytic cleavage of the mRNA near the site of ribosome stalling. However, the requirement for this cleavage event (long taken as fact, and used as a surrogate marker for NGD in numerous studies) has never been rigorously tested because the responsible nuclease was never identified.This study identifies Cue2 as a candidate nuclease through an over-expression screen and shows that its overexpression stimulates NGD substrate degradation, while its deletion abolishes the detection of the endonucleolytic fragment. These and other data are used to argue that Cue2 is the long-sought endonuclease in NGD (although some caveats are warranted here; see point 6 below). Rather remarkably (and presumably very disappointingly), the deletion of Cue2 has no detectable effect on mRNA levels of NGD substrates, strongly arguing that the endonucleolytic fragment is neither a good marker for NGD (probably warranting a re-evaluation of many earlier interpretations) nor a required intermediate in the degradation process. In light of this conclusion, the remainder of the paper presents evidence that there is a parallel pathway for clearance of stalled/collided ribosomes via a helicase called Slh1 (which was already vaguely implicated in these processes by the Inada and Brandman groups in earlier work). However, relatively little can be said about what Slh1 is doing, what it acts on, or how it works. While the study has a number of interesting findings for the field (particularly the striking result that endonucleolytic cleavage is not a substantial pathway of mRNA decay under standard lab yeast conditions), its broader appeal is limited at present because a rigorous demonstration that Cue2 is indeed the nuclease is lacking, and more importantly, the characterization of the alternate pathway of ribosome clearance by Slh1 is overly superficial (highlighted by the rather vague model shown in Figure 3a). For this reason, the study is too preliminary at this stage of analysis for publication in the journal.Specific points:1) The authors state that the GFP signal reflects reporter mRNA levels; however, the GFP signal would also be affected by any effects on translation (especially initiation). This is relevant because ribosome stalling might affect initiation, and effects on translation might be the explanation for some of the tRNA modification ‘hits’ and others. This caveat should be mentioned and taken into account in interpreting any of the screen results where mRNA levels are not directly examined.2) Genes needed for NGD should be evident as genes whose deletion causes an increase in the Log2(GFP/RFP) score for the NGD reporter but not for the OPT control. Yet, the authors do not comment explicitly on these 20 or more hits shown in Figure 1C or Supplementary Figure S1B. I presume none of these proved to be of interest; nevertheless, it is worth briefly mentioning in the text what genes were common to both reporters, and why they were not investigated further as NGD factors.3) Related to point 2, when discussing the candidates on page 5, the authors should be more explicit about whether the hits inhibited or enhanced NGD. For example, they note tRNA modification genes; do these inhibit or enhance NGD, or both? Some precision in the text would be appreciated by most readers, who won’t download the raw supplemental data.4) The effects of Dom34 and Ski2 deletion in Figure 1F are difficult to appreciate. For example, I am not able to see any effects on the Northern or western blot for either gene deletion, and it is unclear whether the very small effect seen with the reporter assay is statistically significant. Were the Northern and western blots quantified from multiple independent experiments to verify that there really is a difference, and if so, what is the magnitude of the difference. Dom34, Hbs1, Ski complex members, and Xrn1 are not indicated in Figure 1C or S1B, but inspection of Table S1 shows that NONE of these were hits by the authors’ definition. Rather strikingly however, the experiment in Figure 2i suggests that Xrn1 deletion has a very strong effect on their NGD-CGA reporter, yet it shows no effect in the screen. From what is presented, it seems the deletion screen was not especially successful in even identifying known factors involved in NGD, including a gene (Xrn1) whose direct and strong effect on NGD should have been very clear. This raises the question of what the ‘hits’ actually mean (see point 1). Regardless, the authors statement that “the identification of these candidate genes provided strong validation of the genetic screen” is not really accurate. As things stand, I cannot see how presentation of the deletion screen as one that is monitoring NGD can be justified.

Points 1-4: While we strongly disagree with the assessment that the screen is not a strong indicator of NGD function, we appreciate many of the comments made by the reviewer. Yes, GFP levels might also be affected by factors that affect translation initiation and we think we found several of these factors including the yeast homologs of the mammalian GIGYF2. The tRNA modification enzymes are also interesting and increase GFP levels; we do not know why, but they would be interesting to characterize.

We argue that because our screen identified essentially all the known factors involved in the upstream steps of these pathways (Asc1, Hel2, and all the components of the RQT complex) and because of the strong overlap in candidates between NGD-AAA and NGDCGA that our screens really were “working” and monitoring NGD. We add that although much work has been done to characterize the role of these known factors in protein output, less work has been done to clarify these factors’ function on a reporter such as our own. As such, it is a priori difficult to predict our ‘hits’ or attempt to explain them without follow up experiments. Furthermore, for example, XRN1 would not be expected to come up in our screen because the XRN1 deletion would merely stabilize decapped mRNA, which will not be translated and thus won’t yield increased GFP signal. Nevertheless, as we now focus more closely on Cue2 and its function in the revised manuscript, we have not included the deletion screens that we will continue to follow up on in our future studies.

5) Cue2 is postulated to act on collided ribosomes when it is recruited via Hel2- dependent ubiquitination to the site of collision. The authors do not directly show that ubiquitin or the CUE domain are needed; however, a key test of the model that Cue2 has any role in NGD is that deletion of Hel2, which mediates collided ribosome ubiquitination, should reduce NGD. While Hel2 deletion does abrogate the 16-mer ribosome footprints in the profiling experiment, Table S1 indicates that Hel2 deletion strongly increases NGD (at least judged by the GFP/RFP ratio). At least according to the screen results, it would seem that Hel2 is not on the pathway to NGD as previously thought, but rather a suppressor of NGD (despite the apparent result that Hel2 is required for Cue2-dependent endonucleolytic cleavage). How do the authors reconcile these seemingly conflicting observations?

Point 5: The reviewer catches an important discrepancy with the literature where indeed Hel2 is typically required for NGD and thus might naively be expected to emerge here as an increase in GFP expression (Brandman et al. Cell, 2012). Indeed, we do show strong evidence that Hel2 is needed for generation of the Cue2-mediated 16 nt RPFs, thus nicely agreeing with earlier studies (Simms et al. Mol Cell, 2017). Our explanation however for the decrease in GFP levels in the HEL2 deletion strain is that ribosomes are known to frame-shift on these mRNAs when not targeted by Hel2, and then are likely targeted for decay by NMD. We are currently testing these ideas, but as HEL2 is not a major focus of our study and because the deletion screen is no longer being included, this is beyond the scope of the revised manuscript.

6) A major deficiency of the study is that the authors cannot rule out the possibility that Cue2 is acting indirectly to influence endonucleolytic cleavage. They present no data to show that Cue2 binds ribosomes, or data to illustrate that recombinant Cue2 has activity against collided ribosomes. Furthermore, the point mutations show incomplete effects, which is not what one would expect if a key catalytic residue was disrupted. If the claim is that Cue2 is the endonuclease that acts on collided ribosomes, it is crucial that the authors minimally show that: (i) Cue2 binds to collided ribosomes (can be easily generated as in PMID 28943311) but not translating monosomes or non-collided polysomes; (ii) recombinant Cue2 has endonuclease activity toward its putative target and that this activity is disrupted by a mutation in the putative catalytic site. In the absence of this, the authors risk making the same mistake recently made for Vms1 – a vague structural similarity to eRF1 leading to a proposed mechanism that turned out to be wrong very soon thereafter.

Point 6 -We appreciate the reviewer’s concerns and have addressed them more completely in our revision as discussed in reply to reviewer #1. An important point to make is that while our mutational analysis is broadly convincing, the effects of single mutations are not complete as might be expected from a metal-independent endonuclease with a “composite” active site composed of Cue2 itself and the surrounding regions of the ribosome/mRNA target. For example, in a recent study by Simms et al. PLOS Genetics, 2018), it is argued that endonucleolytic cleavage depends in part on elements of the ribosome (RPS3).

7) The site of endonucleolytic cleavage relative to the site of stalling has been characterized in some detail in the past by both the Green and Inada groups, and the fact that collided ribosomes behind the lead ribosome is in the rotated state has been established by structural studies from the Hegde/Ramakrishnan and Beckmann/Inada groups. Thus, the ribosome profiling analysis to map Cue2-dependent cleavage sites and sites of trailing ribosomes, while satisfyingly consistent, does not add much to the study. The effects of Slh1 deletion are new, but without any further analysis, these profiling data do not provide much insight into what Slh1 is actually doing.

Point 7: We agree that the cryoEM structures of colliding ribosomes provides much context for interpreting our ribosome profiling data and that our high resolution profiling data is largely confirmatory for the conformational state of colliding ribosomes. We disagree, however, that previous studies have identified the sites of cleavage by Cue2; our data here are new and exciting. It is true that previous studies have contributed data to identifying the sites of cleavage. Our own data argued that the endonuclease was likely to be ribosome associated based on the conserved frame of the cleavage sites (Guydosh and Green RNA, 2017). Other studies suggested that RPS3 might affect endonucleolytic cleavage, thus again associating cleavage events with the ribosome itself (Simms et al. PLOS Genetics, 2018). Other recent mapping efforts (Ikeuchi et al. EMBO J., 2019) argue that a minor cleavage site is found within the rotated ribosomes though these studies did not have the resolution to exactly mark cleavage in the A site. Our data identify a very precise cleavage site in the A site of the ‘colliding’ ribosome. This makes a strong prediction about where Cue2 binds and this prediction nicely correlates with the structural homology modeling arguing that Cue2 could bind in the A site. This is an important and novel contribution.

8) The genome-wide mapping of 16-mer footprints in Cue2 deletion strains is enriched in prematurely polyadenylated transcripts. While these data support a ‘broad role’ in the sense that many such transcripts are detected, this is probably overstating the result. The profiling data cannot be used to easily infer the abundance or proportion of any message that produces the 16-mer products. Thus, it is entirely possible from what is shown in the paper that the change in 16-mer to 21-mer ratio that is plotted for any given message might reflect the fact that 16-mers originating from a very small proportion of that message (say 2%) increases 2-3 fold (to say 4-6%). While this might occur on many messages, it would not be an appreciable contributor to degradation. This seems likely given the authors’ data that even engineered and robustly-stalling NGD reporter substrates are not impacted detectably in their degradation by deletion of Cue2. Unless they can produce data showing that at least some of these putative endogenous substrates of NGD are actually stabilized appreciably in Cue2 deletion strains, a more cautious interpretation of the profiling data is warranted.

Point 8: The reviewer is correct on these points. The mRNA targets that we identify as enriched in Cue2-dependent 16 nt RPFs are likely predominantly decayed by the more canonical Xrn1- or exosome dependent pathways and Cue2 activity may account for relatively little in terms of overall mRNA levels. Indeed, we directly show that the Cue2 pathway is normally a relatively minor one for QC on these mRNAs and we provide strong evidence that Xrn1 is the major mediator of decay for NGD-CGA (and thus likely for other cellular targets). That is the biology of the cell – and we have defined it – this is an important contribution to our understanding as a field. We counter that the strong conservation of Cue2 across eukaryotes argues for an important role for this pathway across biology and that a next step will be to understand when and where the pathway is most critical. Our genetic observations with Slh1 suggest that when the normal QC pathways are overwhelmed, that Cue2 may indeed play a more essential role.

9) Minor point: have the authors verified that Cue2 overexpression reduces the mRNA level of the NGD-AAA reporter similar to that seen with NGD-CGA in Figure 1G?

Point 9: For simplicity, we focused our experiments on the NGD-CGA reporter. Since we also show Cue2-dependent endogenous cleavages on known prematurely polyadenylated substrates, we did not think that showing additional follow up experiments on the NGD-AAA reporter added very much.

10) Minor point: Figure 3B does not have WT, so a reader cannot judge that the Dom34/Ski2 double deletion is indeed more sensitive to CHX. If available, please add this control (I realise the comparison is shown in Supplementary Figure S4 but many readers outside the field won’t bother to scrutinize the supplement).

Point 10: These spotting assays were problematic for multiple reviewers and we agree. We have dropped this assay from the revised manuscript.

11) Minor point: The authors may wish to cite the recent Simms et al. study in PLOS Genetics where they show that mutations to RPS3 can abolish the endonucleolytic cleavage fragment. Consistent with the Cue2 deletion data in the current study, Simms et al. find essentially no effect on the degradation rate of a NGD substrate, again arguing that endonucleolytic cleavage is not a requisite intermediate in mRNA degradation.

Point 11: The reviewer is correct - we have included this additional citation.

Reviewer 3:

D’Orazio et al. set up a thoughtful screen to identify NGD factors distinguished from RQC protein degradation factors. As a result of knock out and overexpression library screening, they identified Cue2 and Slh1 parallel pathways to resolve stalled ribosomes on problematic mRNAs. Their results suggest that Cue2 is the endonuclease that triggers the NGD. They propose that endonucleolytic cleavage is dependent on Cue2.The NGD endonuclease was a mystery in the RNA field for a while, but simply giving a name to the endonuclease falls short for the expected impact for a broad audience. No concrete molecular mechanisms are provided. For example, how is Cue2 recruited to problematic mRNAs? In fact, it is unfortunate that no direct evidence that Cue2 is even an endonuclease is provided as the data is based largely on correlations and does not directly look for this activity either biochemically or within a reconstitution system. Most of the studies are highly dependent on reporters and the effects observed are only upon the deletion of multiple pathway components or reporter overexpression. Therefore, the study does not provide enough major mechanistic details and there is a concern that the impact of these pathways is not adequately defined in a physiologically meaningful manner. Moreover, I am not convinced by the data the authors offer. The evidence is preliminary and partial.Major limitations of the study include:1) Superimposition of a homology model of Cue2 using a bacterial translation initiation factor doesn't prove anything (Figure 2). First of all, bacterial and eukaryotic ribosomes have very different structures. Even if both Cue2 and IF3 have an SMR domain, these are totally different proteins and have distinct working contexts. Of course, IF3 is close to mRNA at the A-site to initiate translation, but we never know where Cue2 is or even direct evidence that it associates with the ribosome. If the authors want to discuss the position of Cue2 SMR domain on the ribosome, they need to solve the structure using eukaryotic ribosome and Cue2 protein.

Point 1: The reviewer is correct. We do not have a structure of Cue2 on the ribosome nor do we indicate that Figure 2 and S2 (the structural panels) represent anything more than structural homology modeling. Given that we believe that we have identified the endonuclease and the site of cleavage within the A site of the colliding ribosome (from profiling data) we think this structural homology modeling provides exciting insights into Cue2 function.

2) No statistical information is present at all in this manuscript. Even if it is obvious to be significant, it is necessary to include this information. All the graphs need to have statistical information such as FDR or p-values including Figure 1B, C, D, F, G, Figure 2F, Figure 4F. It is essential to have statistics for deep sequencing data for the entirety of Figure 4 and 5. Without statistics, no specific or genome-wide conclusions can be made.

Point 2: We have performed all experiments multiple times and now present all data to show the level of significance (including flow cytometry data, northerns and ribosome profiling experiments).

3) The impact of these pathways for biology remains unclear. They show correlations with cryptic polyA genes without statistics and provide two genes, but these cells grow normally in the conditions that the Ribo-seq library was produced according to Figure S4, comparing WT and dom34#ski2#. There is no validation of these preliminary Ribo-seq experiments. For example, the authors could have easily tagged endogenous RNA14 and Yap1 and looked at protein/transcript levels. In Figure 5 legends it is stated, “genes below the diagonal correspond to those with decreased ratios of 16 nt RPFs”. I disagree with this conclusion as without replicates and careful analysis a lot of these mRNAs could just be noise.

Point 3: We now provide replicates on all profiling experiments and show by DE-Seq that the mRNA targets that we identify are statistically significant. We have chosen not to follow the protein output or transcript levels of specific mRNA targets for multiple reasons: first, under normal conditions, we know that Cue2 is a minor pathway that contributes to the loss in stability of such prematurely polyadenylated mRNAs and second, because the amount of premature polyadenylation that takes place on any transcript is likely to be relatively small. Nevertheless, we believe these pathways exist because even small amounts of such dysfunctional mRNAs can generate problematic amounts of toxic peptide products. Also, see above – Reviewer 2, point 8.

4) The experiments in Figure 3 where the authors attempt to address their model genetically are interesting. By treating with low-doses of cycloheximide, they assume that the observed effects are related to the effects of this treatment on “wide-spread stress on the ribosome quality control machinery”. However, it is entirely possible that Slh1 and Cue2 could have additional or alternative functions and that under the effects of this broad translation elongation inhibitor do not directly relate to clearance of stalled ribosomes or NGD. They need to explain the phenotypes observed. Where does the toxicity come from? The authors should have carried out Ribo-seq from these genetic backgrounds upon cycloheximide treatment and found the targets relevant for the toxicity.

Point 4: We appreciate the reviewer’s concerns with the use of CHX treatment as a proxy for translational distress. We argue however that this particular stress is used throughout the field in this manner and that there exists considerable data to indicate that light CHX (or other antibiotics that target translation) treatment does indeed lead to genome wide ribosome collisions (Simms et al. Mol Cell 2017; Juszkiewicz Mol Cell 2018) (also see new biochemistry in revised Figure 5). We do not believe that an RNA-seq experiment on CHX-treated cells would yield critical information on the mechanisms of toxicity. Again, we have dropped this assay since it is no longer clarifying for the manuscript.

5) Since the Cue2 pathway is the minor pathway, it could be possible that the effects of the Cue2 pathway are observed only when it is forced, such as upon deletion of multiple genes or reporter overexpression and the Slh1 pathway may have more of a physiological meaning. However, the molecular mechanisms of the major Slh1 pathway are unsolved. The authors provide a model that the Slh1 helicase takes out stalled ribosomes from mRNA, but they don't provide any direct evidence into how this is achieved. Since this is the major pathway for problematic mRNAs, they need to define the molecular mechanism of this pathway and what the relevant mRNAs are as they pertain to Figure 3B.

Point 5: The reviewer is correct. We have not firmly established that Slh1 functions to remove colliding ribosomes from problematic mRNAs. We have, however, provided several key new pieces of evidence on Slh1 function. First, we show that Slh1 function negatively regulates Cue2 function, likely preventing assembly of a Cue2-competent substrate or competing directly with its function. We support these ideas both with northern analyses and with ribosome profiling data. Second, we show by ribosome profiling that ribosome occupancy (RO) is altered by Slh1 function (deletion of SLH1 leads to increased RO on the reporter mRNA). Nevertheless, we have no in vitro biochemistry per se to support any particular model of action for Slh1. Our model now is deliberately vague on this point (Figure 7) and simply places SLH1 genetically to indicate that its action regulates that of CUE2; these ideas are markedly consistent with other data in the field. We hope this more conservative model is helpful in framing our observations.

6) Cue2 has not been directly characterized as an endonuclease by the authors. They provide information about the RNase activity of other SMR domain containing proteins but the endonuclease activity of Cue2 is never directly biochemically tested, for example in a reconstitution system.

We appreciate this concern and now include direct in vitro biochemical evidence as well as a mutational analysis of Cue2 at its endogenous location in the genome. See Reviewer 1, point 1.

Minor points:1) Why is the Flag signal increased in the slh1# strain in Figure 1F? Is that because of a defect in RQC or increased ribosome read through?

Slh1 has been shown to prevent ribosome read though and this is a full-length product; therefore we believe this results from ribosome read-through and a defect in RQC (i.e. these are coupled activities).

2) In Figure 2I why does xrn1# increase the levels of the full-length RNA? Is this dependent on the Slh1 pathway? If that is the case, does the Slh1 pathway somehow selectively accelerate RNA turnover of problematic RNAs. How?

Our northern blots suggest that Xrn1-mediated decay of problematic mRNAs is not wholly dependent on Slh1. The reviewer rightly points out that there are some interesting questions here to address in order to figure out the many ways that problematic mRNA recruit the different decay machineries. These are questions that we plan to address in future work.

3) In Fig4ure D, what are the two highest peaks in the 16nt RPFs in the slh1# dom34# strain? These are shifted away from cleavage sites.

We believe these are cleavage sites between colliding ribosomes as previously characterized both in our own work (Guydosh and Green, RNA 2017) and by Inada and colleagues (Ikeuchi et al Nat Comm 2019). These data suggest to us that there may be another endonuclease (and other data identifying multiple ubiquitination events that lead to endonucleolytic cleavage, Ikeuchi et al. 2019). Interestingly, essentially all 16 nt RPFs disappear on deletion of CUE2 suggesting minimally that CUE2 is essential for all cleavage events to occur. There is more work to be done in characterizing these secondary cleavage sites.

4) I believe that using * instead of ski2# is misleading. They should clearly write ski2# background in Figures 3 and 4 as well as in the legend.

Yes. This was a mistake (though meant to simplify) – we have corrected this in our revised version.